



Atmospheric
Chemistry
and Physics

Research article

# Volatile organic compound fluxes over a winter wheat field by PTR-Qi-TOF-MS and eddy covariance

Benjamin Loubet[1], Pauline Buysse[1], Lais Gonzaga-Gomez[1], Florence Lafouge[1], Raluca Ciuraru[1], Céline Decuq[1], Julien Kammer[1,a], Sandy Bsaibes[1,2], Christophe Boissard[2,3], Brigitte Durand[1], Jean-Christophe Gueudet[1], Olivier Fanucci[1], Olivier Zurfluh[1], Letizia Abis[1,b], Nora Zannoni[2,c], François Truong[2], Dominique Baisnée[2], Roland-Sarda Esteve[2], Michael Staudt[4], and Valérie Gros[2]

[1]UMR ECOSYS, INRAE, AgroParisTech, Université Paris-Saclay, 78850, Thiverval-Grignon, France
[2]Laboratoire des Sciences du Climat et de l'Environnement, LSCE, UMR CNRS-CEA-UVSQ, IPSL, 91191 Gif-sur-Yvette, Île-de-France, France
[3]Université de Paris and Univ. Paris Est Creteil, CNRS, LISA, 75013 Paris, France
[4]CEFE, CNRS, EPHE, IRD, Univ Montpellier, Montpellier, France
[a]now at: Aix Marseille Univ., CNRS, LCE, Marseille, France
[b]now at: Umweltchemie und Luftreinhaltung, Technische Universität Berlin, Straße des 17. Juni 135, Berlin, 10623, Germany
[c]now at: Max-Planck Institute for Chemistry, Hahn-Meitner-Weg 1, 55128 Mainz, Germany

**Correspondence:** Benjamin Loubet (benjamin.loubet@inrae.fr)

**Abstract.** Volatile organic compounds (VOCs) contribute to air pollution through the formation of secondary aerosols and ozone and extend the lifetime of methane in the atmosphere. Tropospheric VOCs originate to 90 % from biogenic sources on a global scale, mainly from forests. Crops are also a potentially large yet poorly characterized source of VOCs (30 % of the VOC emissions in Europe, mostly oxygenated). In this study, we investigated VOC fluxes over a winter wheat field by eddy covariance using a PTR-Qi-TOF-MS with high sensitivity and mass resolution. The study took place near Paris over a 5-week period and included flowering, crop maturity and senescence. We found a total of 123 VOCs with fluxes 3 times above the detection limit. Methanol was the most emitted compound with an average flux of $63\,\mu\mathrm{g\,m^{-2}\,h^{-1}}$, representing about 52 % of summed VOC emissions on a molar basis (36 % on a mass basis). We also identified ethanol, acetone, acetaldehyde and dimethyl sulfide among the six most emitted compounds. The third most emitted VOC corresponded to the ion $m/z$ 93.033. It was tentatively identified as furan ($C_6H_4O$), a compound not previously reported to be strongly emitted by crops. The average summed VOC emissions were about $173 \pm 6\,\mu\mathrm{g\,m^2\,h^{-1}}$, while the average VOC depositions were about $109 \pm 2\,\mu\mathrm{g\,m^{-2}\,h^{-1}}$ and hence 63 % of the VOC emissions on a mass basis. The net ecosystem flux of VOCs was an emission of $64 \pm 6\,\mu\mathrm{g\,m^{-2}\,h^{-1}}$ ($0.5 \pm 0.05\,\mathrm{nmol\,m^{-2}\,s^{-1}}$). The most deposited VOCs were identified as hydroxyacetone, acetic acid and fragments of oxidized VOCs. Overall, our results reveal that wheat fields represent a non-negligible source and sink of VOCs to be considered in regional VOC budgets and underline the usefulness and limitations of eddy covariance measurements with a PTR-Qi-TOF-MS.

## 1 Introduction

Volatile organic compounds (VOCs) are key compounds for atmospheric chemistry that contribute to the production of harmful pollutants to human health, among which are ozone ($O_3$) and secondary organic aerosols (SOAs) (Monks et al., 2015; Lang-Yona et al., 2010). Ozone, which is also a powerful greenhouse gas (IPCC, 2018), affects vegetation growth with an estimated annual cost of EUR 11–18 billion on agricultural production worldwide in the last decade (Ashmore, 2005; Avnery et al., 2011). Similarly, particulate matter (PM) including SOA directly and indirectly affects global warming by modifying radiation scattering and the cloud albedo (Makkonen et al., 2012). The official legal directives on VOC emissions are not very restrictive. The main regulations in Europe, resulting from the Gothenburg Protocol of 1999, concern the limitation of their emissions from industrial plants and the limitation of their concentrations in consumer products. The emission of fine particles ($PM_{2.5}$), on the other hand, is more tightly regulated. European countries must meet air pollution targets set out in the European Air Quality Directive (2008/50/EC TS1). Since VOCs are precursors of harmful air pollutants, their sources need to be better quantified to identify potential remedies to mitigate $PM_{2.5}$ and ozone threats.

Of the $760 \, \mathrm{Tg} \, (\mathrm{C}) \, \mathrm{yr}^{-1}$ VOC emitted globally, it is estimated that around 90 % have a biogenic origin (BVOC) (Sindelarova et al., 2014). Forests are the main emitters and isoprene is the most emitted compound worldwide. In Europe, managed ecosystems (crops, managed grasslands and forests) representing about 50 % of the land area are the largest VOC source in the continent. According to the European BVOC inventory by M. Karl et al. (2009), forests account for 55 % of the total emission, agricultural lands for 27 %, and grasslands, wetlands and shrubs for 18 %. These values are however based on few datasets of BVOC fluxes (Keenan et al., 2009), especially for crops. This leads to uncertainties in the estimates of BVOC fluxes from this type of ecosystem. To date, only a few studies have investigated VOC fluxes from crops, most of them by chamber techniques (Copeland et al., 2012; Crespo et al., 2013; Eller et al., 2011; Konig et al., 1995; Das et al., 2003; Graus et al., 2013; Gonzaga Gomez et al., 2019; Wiss et al., 2017). Flux measurements made at the canopy level on crops are still rare and have been rather focused on orchards and grasslands (Karl et al., 2001, 2005; Copeland et al., 2012; Fares et al., 2012; Miresmailli et al., 2013; Misztal et al., 2014; Bachy et al., 2016; Brilli et al., 2016; Zenone et al., 2016; Berhongaray et al., 2017; Portillo-Estrada et al., 2018; Bachy et al., 2018, 2020; Gallagher et al., 2000). Bachy et al. (2016) reported fluxes of isoprene, monoterpenes, methanol, acetone, acetaldehyde and acetic acid from a maize field, which revealed lower emissions than those parameterized in VOCs emission models like MEGAN or ORCHIDEE (Messina et al., 2016). In a follow-up study, Bachy et al. (2020) reported

the first measurements of VOC fluxes above a winter wheat field by eddy covariance with a quadrupole proton transfer mass spectrometer (PTR-MS). More flux measurements have been reported above grasslands than above crops (e.g. Custer and Schade, 2007; Bamberger et al., 2010; Müller et al., 2010; Ruuskanen et al., 2011).

Reported VOC emission rates from crops are variable over a wide range of values as shown by Gonzaga Gomez et al. (2019) and Bachy et al. (2016). Thus, more data are needed to obtain reliable regional emission estimates. Studies measuring ecosystem-scale fluxes are particularly useful, since they integrate VOC sources and sinks throughout the canopy. For example, Bachy et al. (2018) showed that methanol can be emitted from bare agricultural soils in comparable quantities to plants. The emergence of highly sensitive time of flight PTR-MS (PTR-TOF-MS) (Sulzer et al., 2014) enables the detection of a lot more VOCs than with previous quadrupole PTR-MS and allows ecosystem-scale measurement of their fluxes by eddy covariance. This opens the possibility of obtaining a much more complete spectrum of VOC fluxes from crops than in previous studies. The objective of this study was therefore to quantify the fluxes of VOC exchanged between a winter wheat field and the atmosphere at the ecosystem scale using the eddy covariance method over periods of flowering, grain filling and senescence, using a highly sensitive PTR-Qi-TOF-MS. The methodology to compute the fluxes and their uncertainties are presented in detail, and the emission and deposition fluxes are discussed in terms of their magnitude and timing.

## 2 Methods

### 2.1 Experimental site and crop management

Flux measurements took place at the Integrated Carbon Observation System (ICOS) FR-Gri site (Grignon, 48°51′ N, 1°58′ E) located about 30 km west of Paris (France), over a 46 d period between the 3 June and the 19 July 2016. The site is a 19 ha field (Fig. 1), with a crop rotation of wheat, maize, barley and occasionally rapeseed and with a winter cover crop before maize. The soil is classified as Luvisol consisting of 25 % clay, 70 % silt and 5 % sand. More details about the site can be found in Loubet et al. (2011). The site that is part of a dairy farm receives a lot of nitrogen as mineral or organic matter, which leads to large ammonia volatilization to the atmosphere (Personne et al., 2015). The field was also shown to be a source of NO (Vuolo et al., 2017) and HONO (Laufs et al., 2017). The field is around 450 m downwind from the farm buildings, with about 250 dairy cows and 900 lambs. The farm has a storage tank for manure and storage areas. The farm buildings were shown to be a large source of ammonia that can be detected from the field site (Loubet et al., 2012). Similarly, Kammer et al. (2020) demonstrated that the animal farm is a consistent source of VOC and especially methanol, ethanol and acetaldehyde but

also a specific source of trimethylamine and dimethylsulfide (DMS) that might hence be detected at the field site.

Winter wheat was sown on 20 October 2015 at a density of $2\,500\,000$ plant ha$^{-1}$, as a mixture of three varieties (Rubisco, Atlass and Premio). The crop was fertilized four times with a nitrogen solution ($25\,\%$ NO$_3$, $25\,\%$, NH$_4$, $50\,\%$ urea) at a rate of 84, 39, 39 and $55\,kg\,N\,ha^{-1}$ on 1 March 2016, 9 April 2016, 29 April 2016 and 10 May 2016. Following the emergence of brown rust or *Septoria*, the crop was sprayed with a fungicide on 13 April 2016 (Cherokee; cyproconazole g L$^{-1}$, propiconazole $62.5\,g\,L^{-1}$, chlorothalonil $375\,g\,L^{-1}$, at a rate of $1.25\,L\,ha^{-1}$), 16 May 2016 (bixafen $75\,g\,L^{-1}$ and prothioconazole $150\,g\,L^{-1}$ at a rate of $0.7\,L\,ha^{-1}$) and 4 June 2016 ($250\,g\,L^{-1}$ azoxystrobine at a rate of $0.27\,L\,ha^{-1}$). The crop was harvested on 28 July 2016 with quite a small yield of $4.3 \times 10^3\,kg\,DW\,ha^{-1}$ (DW: dry weight). The crop started flowering on 20 May and was fully flowered on 1 June 2016. On 9 March and 28 July 2016, aboveground biomass consisted of $13\,\%$ and $0\,\%$ green leaves, $4\,\%$ and $14\,\%$ yellow leaves, $65\,\%$ and $45\,\%$ stems, and $17\,\%$ and $41\,\%$ grain ears, respectively. The maximum leaf area index (LAI) of green leaves, indicative of the photosynthesis activity, was $7.3\,m^2\,m^{-2}$ on 25 May and decreased to $3.4\,m^2\,m^{-2}$ on 9 June. Finally, the canopy height varied from 0.9 to 1.1 m during the experiment with quite large variability inside the field (0.15 m standard deviation).

## 2.2 Volatile organic compound measurements

### 2.2.1 VOC eddy covariance sampling system

VOCs were measured in 30 min cycles. The eddy covariance flux was recorded over 20 min, while the last 10 min were devoted to vertical profiles (5 min) and chamber (3 min) mixing ratio measurements and zero checks (2 min) by sampling through a hydrocarbon filter. In this study, the profiles and chamber measurements are not presented. A 16-way Sulfinert coated valve, located 5 cm from the drift tube of the PTR-Qi-TOF-MS and heated to 80 °C, was used to switch between the eddy covariance and other channels. All sampling tubes were made of Teflon PFA (perfluoroalkoxy) surrounded by heating tape inserted in an insulator. Tubes were constantly heated to 60 °C using a homemade thermostatted device and connected to the PTR-MS inlets heated to 80 °C. A stainless-steel manifold made of Swagelok T fittings was used to subsample the air from the covariance line to the PTR-MS and other instruments. The residence time in the covariance line averaged 2.15 s, which consisted of 1/2 in. tubing (external) flushed with a Busch SV-1010 pump (Busch, Switzerland) with an air flow rate of $42\,L\,min^{-1}$. This high flow rate has been set to minimize any chemical reaction and ensure a turbulent flow (Reynolds number $\sim 6000$) to keep the high frequency of concentration fluctuations for the eddy covariance method. The end pressure in the line was 240 mbar below

ambient pressure. Air from the covariance line was drawn to the 16-way valve at a rate of $0.5\,L\,min^{-1}$ via a 1 m long, 1/8 in. external diameter PFA tubing heated to 80 °C.

The VOC eddy covariance mast comprised a sonic anemometer (R3-50, Gill, UK) placed at the reference height of 2.7 m aboveground and an open-path IRGA (infrared gas analyser) for CO$_2$ and H$_2$O measurement (Li-7500, LI-COR, USA) placed 0.2 m away to the east and 0.1 m below the anemometer centre (Fig. 2). The covariance line head that sampled air to the PTR-Qi-TOF-MS was placed 0.1 m to the east and 0.2 m below the anemometer centre. The eddy covariance line head was made of a stainless-steel cup ($\sim 2.5$ cm diameter with a 1 mm mesh Teflon screen). The covariance line was used for VOC fluxes and mixing ratio measurements as well as for CH$_4$, N$_2$O, H$_2$O, NO$_x$ and O$_3$ mixing ratios. Ozone and NO$_x$ analysers required a teflonized (internal chambers in Teflon) subsampling pump (KNF laboport N820, France) to increase the pressure to ambient.

### 2.2.2 PTR-Qi-TOF-MS instrument setup

A PTR-Qi-TOF-MS (Ionicon Analytik GmbH, Innsbruck, Austria) was used for continuous online measurements of VOC mixing ratios at 10 Hz. The analyser has been described in detail before (Abis et al., 2018; Sulzer et al., 2014). The drift tube was maintained at $3.5 \pm 0.001$ mbar and a temperature of $80 \pm 0.16$ °C, while the drift voltage $E$ was set to $995 \pm 0.8$ V before the 29 June and to $848 \pm 0.3$ V after. The corresponding $E / N$ ratios (where $N$ is the gas density) were $150.66 \pm 0.15$ Td and $129.03 \pm 0.21$ Td (1 Townsend $= 10^{-17}$ V cm$^{-2}$). The $E / N$ ratio at the start of the experiment was rather high and was hence lowered down to 129 Td in the second half of the experiment to minimize cluster formation and fragmentation, although the latter cannot be completely avoided (Pang, 2015; Tani et al., 2003). The number of detection channels was set to $240\,000$ and the mass spectrum spanned $m/z$ 15 to $m/z$ 530. The extraction rate of ions in the TOF was set to 40 µs (25 kHz), meaning that 2500 extracted spectra were averaged before being recorded at 10 Hz. The mass-resolving power corresponded to a resolution (ratio of ion peak width at mid-height to peak value) of around 4500 during the experiment. This means that the instrument had a mass-resolving power of $\sim 0.007\,m/z$ at $m/z = 30$ and $\sim 0.03\,m/z$ at $m/z = 150$.

### 2.2.3 Acquisition and pre-processing of PTR-Qi-TOF-MS data

A data acquisition software was developed on Labview to synchronize the measurements of the PTR-Qi-TOF-MS with the sonic anemometer and the other fast-response instruments and to store the data. This software was based on the acquisition of a list of ion peaks integrated online by the TOFDaq software (TOFWERK, SW) using shared variables that are exchanged between the acquisition computer and the

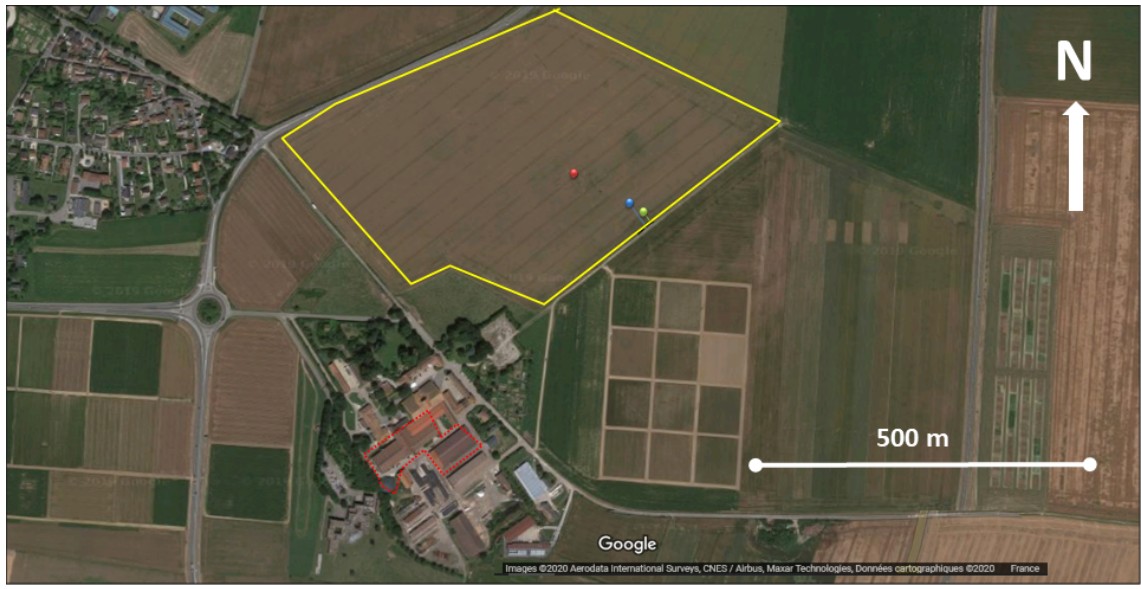

**Figure 1.** Satellite image of the site showing the ICOS FR-GRI field (yellow line), the ICOS flux station (red dot) the VOC eddy covariance sampling site (blue dot) and the VOC profile sampling site (green dot). The dotted red line shows the farm buildings where animals are mainly stored, on the southeast of the field. © Google Earth.

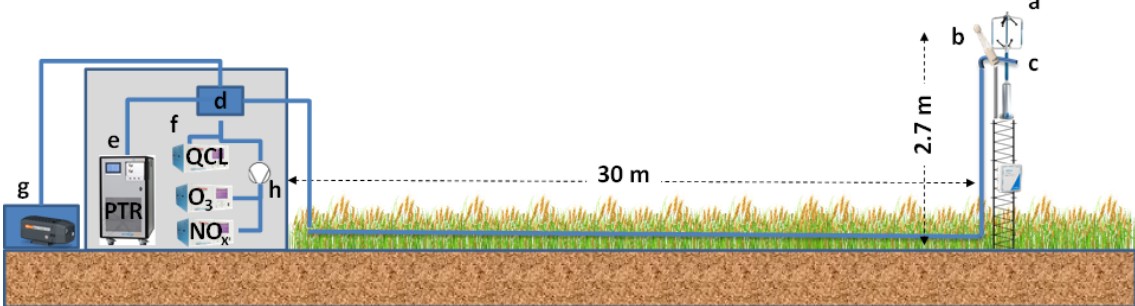

**Figure 2.** The eddy covariance VOC measurement setup, showing (**a**) the ultrasonic anemometer, (**b**) the $CO_2$ and $H_2O$ open-path analyser, (**c**) the heated sampling line inlet, (**d**) the heated manifold, (**e**) the PTR-Qi-TOF-MS, (**f**) the gas analysers, (**g**) the high-volume vacuum pump, and (**h**) the teflonized subsampling pump. Diagram not to scale.

PTR-Qi-TOF-MS computer via a local Ethernet connexion. The desired ion peak list was set up at the start of the experiment based on a list of known compounds tuned to the site. The acquired ion peak integral counts per seconds (cps) were synchronized at 20 Hz with the ultrasonic anemometer. Every 5 min, a mass calibration was performed by the TOFDaq software using masses $m/z$ 21.0221 ($H_3^{18}O^+$) and $m/z$ 59.049 (acetone $C_3H_6OH^+$). When the calibration of these masses shifted from the real masses by more than 0.005 $m/z$ the 5 min record was discarded. Additionally, the quality of the peak integration was verified a posteriori by integrating several spectra with PTR-MS viewer (Ionicon, V3.3). We found errors lower than 5 % for most of the compounds in the most emitted ions as shown by Gonzaga Gomez et al. (2019). The synchronized data, stored every 5 min in binary files, were pre-processed with another Labview® program to compute the mixing ratios and fluxes.

### 2.2.4 VOC mixing ratio computation

The pre-processing steps provided averages and standard deviation counts per seconds (cps) for the ion peaks selected $R_i$. The mixing ratio of the compound $\chi_{i,\text{ptr}}$ (in ppb) prior to any calibration, was calculated as TS2

$$\chi_{i,\text{ptr}} = 1.657\,e^{-11} \times \frac{U_\text{drift}\,T_\text{drift}^2}{k\,p_\text{drift}^2}$$

$$\times \left( \frac{\text{cps}_{R_iH^+}^\text{trans}}{\text{cps}_{H_3O^+}^\text{trans} + \text{cps}_{H_2O\cdot H_3O^+}^\text{trans}} \right), \tag{1}$$

Atmos. Chem. Phys., 22, 1–26, 2022      https://doi.org/10.5194/acp-22-1-2022

$$cps_{R_iH^+}^{trans} = \frac{TR_{H_3O^+}}{TR_{R_iH^+}} \times cps_{R_iH^+}, \qquad (2)$$

where $U_{drift}$ is the voltage of the drift tube ($V$), $T_{drift}$ is the drift tube temperature in kelvin (K), $cps_{R_iH^+}$ is the cps of the product ion $i$, $cps_{H_3O^+}$ and $cps_{H_2O \cdot H_3O^+}$ are the cps of the ion source and the first water cluster, k is the proton transfer reaction rate assumed to be constant for all compounds ($2.5 \times 10^{-9}$ cm$^3$ s$^{-1}$), trans stands for normalized for transmission, $TR_{H_3O^+}$ is the transmission factor for $H_3O^+$, $TR_{R_iH^+}$ is the transmission factor for the product ion $i$, and $p_{drift}$ is the pressure in the drift (mbar). The transmission curve from the supplier was used to compute the transmission. $cps_{H_3O^+}^{trans}$ was computed from ion $m/z$ 21.022 ($H_3^{18}O^+$) by multiplying by the isotopic factor of $O^{18}/O^{16}$ in water 487.56, taking the first water cluster as the ion peak $m/z$ 37.028. The constant $1.657e^{-11}$ was derived from the PTR-MS geometry (Supplement).

### 2.2.5 Calibration procedures and uncertainties

The PTR-Qi-TOF-MS calibration factor was measured five times during the experiment. To that purpose, we used a standard calibration mixture cylinder containing 102 ppb of benzene, 104 ppb of toluene, 130 ppb of ethylbenzene and 336 ppb of xylene (122 ppb ortho, 121 ppb meta, 123 ppb para; Messer). The gas mixture from this cylinder was diluted with synthetic air (Alphagaz 1, Air Liquide, France), filtered with a hydrocarbon and humidity filter (Filter Super Clean, final purity = 99.9999 %, Restek) and a hydrocarbon trap (Supelco, Supelpure HC). Two fluorinert coated mass flow controllers (Bronkhorst) were used for dilution to generate mixing ratios from zero to 50 ppb. Fluorinert coating was used to minimize wall effects in the mass flow controller.

The background mixing ratio was determined during each calibration using synthetic air passed through a hydrocarbon filter (Supelco ref 22445-12) for 2 min and keeping the last 30 s of the record. The background was also determined every 30 min by passing ambient air through the same filter to account for the ambient humidity in the zero calibration. The background mixing ratio was determined as the minimum between a 10 d moving minimum of the zero measured every 30 min and the zero measured during the calibrations. This procedure was used since the zero air concentration was sometimes much larger than the one measured with filtered ambient air. The zero was then withdrawn from the uncalibrated mixing ratio, providing the zero corrected mixing ratio $\chi_{i,ptr}^* = \chi_{i,ptr} - \chi_{i,ptr}^{zero\ air}$. Eventually, a calibration factor $S_i$ was applied to $\chi_{i,ptr}^*$ to compute the calibrated mixing ratio $\chi_i$, as follows:

$$\chi_i = S_i(t) \times \chi_{i,ptr}^* = S_{toluene}(t) \times \frac{S_i(t_0)}{S_{toluene}(t_0)}$$
$$\times \left( \chi_{i,ptr} - \chi_{i,ptr}^{zero\ air} \right), \qquad (3)$$

where $S_{toluene}(t)$ was computed as the slope of the regression (with intercept forced to zero) between $\chi_{i,ptr}^*$ and the prescribed mixing ratio during calibrations was adjusted in time based on the five calibrations made during the course of the experiment as well as on $E/N$ and MCP adjustments (see Table S1). For the known VOCs, we made a one-off calibration at the start of the experiment on 31 May 2016. Individual calibration factors $S_i(t_0)$ were determined for methanol, acetonitrile, acetaldehyde, ethanol, acroleine, acetone, isoprene, crotonaldehyde, 2butanone, benzene, toluene, o_xylene, chlorobenzene, a_pinen and 1_2_ dichlorobenzene (Table S2). In addition, the calibrations factors reported by Koss et al. (2018) were used to compute calibration factors for 145 more compounds (Tables S2). They did indeed compute a calibration dataset for a similar, though not exactly identical, instrument as used in this study. However, they operated their PTR-TOF-MS instrument with an $E/N$ of 120 Td, which is similar to our setup after the 29 June.

The difference in $S_i(t_0)/S_{toluene}(t_0)$ between Koss et al. (2018) and this study was below 10 % for acetaldehyde, acroleine, acetone and benzene and below 50 % for methanol, acetonitrile and crotonaldehyde but only $< 84$ % for ethanol, isoprene and monoterpenes. It should be noted that in all cases, $S_i(t_0)/S_{toluene}(t_0) - 1$ has the same sign in both studies. Hence, the calibration factors adopted from that study were good for 4 of 11 compounds and fair for more than half of them. For water vapour, the calibration factor and zero were computed hourly using a linear regression between $\chi_{H_2O \cdot H_2O \cdot H+, ptr}$ TS3 and $\chi_{H_2O, Li7500}$.

The relative uncertainty of the calibration factors $\frac{\delta S_i}{S_i}$ was calculated as the sum of relative errors on the linear regressions between prescribed and measured mixing ratios and that of the standard used for calibration. It averaged 12 % over the measured VOCs and was equal to 14.4 % for methanol and 24.3 % for ethanol. For the calibration factors taken from Koss et al. (2018), the reported relative uncertainty was used. Overall the relative uncertainty of $\chi_i$ was computed following error propagation rules from Eq. (3):

$$\frac{\delta \chi_i}{\chi_i}(t) = \frac{\delta S_{toluene}}{S_{toluene}}(t) + \frac{\delta S_i}{S_i}(t_0), \qquad (4)$$

where $\delta$ denotes an uncertainty. The uncertainty in $\chi_{i,ptr}$ was not considered here as it was assumed to be included in $\delta S_i$. The uncertainty of zero air was also neglected compared to the other sources of uncertainty. The relative uncertainty in $S_{toluene}$ is reported in Table S1 and that of $S_i$ in Table S2 (Supplement).

### 2.2.6 VOC eddy covariance fluxes computation

The fluxes were computed as the covariance between the vertical component of the wind velocity and the mixing ratio in dry air $\chi_{i,d}$, neglecting the Webb–Pearman–Leuning density correction terms (Leuning, 2007). Indeed, we assumed that the drift temperature was not co-varying with wind velocity,

which is a sensible assumption given that the PTR chamber is temperature controlled and given the high thermal mass of the PTR chamber compared to the air mass flowing into the instrument every minute. In these circumstances, the PTR chamber temperature was assumed to be independent of the air temperature, and the flux was computed as

$$F_i = \frac{\overline{p_a^d}}{R\overline{T_a}}\overline{w'\chi'_{i,d}},$$ (5)

where $F_i$ is the flux (nmol m$^{-2}$ s$^{-1}$), $\chi_{i,d}$ is the compound mixing ratio in dry air measured by the PTR (ppb), $w$ is the vertical wind component, $\overline{T_a}$ is the air temperature (K), $\overline{p_a^d}$ the dry air pressure (Pa) and $R$ the ideal gas constant (8.31 J mol$^{-1}$ K$^{-1}$). Overbars denote averages and primes denote fluctuations following the Reynolds decomposition. Here $w'$ was calculated by applying two rotations following Aubinet et al. (2000). The covariance between $\chi'_{i,d}$ and $w'$ was calculated after dephasing the two signals with a lag time $\tau$ computed as the time at which the correlation function $\overline{w'(t)\chi'_{i,d}(t-\tau)}$ was the largest in absolute value (Fig. S4). The lag time was set equal for all compounds as the average of the lag time computed for the water cluster, methanol and acetone, which showed very consistent correlation functions between 2.0 and 2.5 s. To compute the covariance, the lag time was constrained between 2.0 and 2.3 s with a fixed value of 2.15 s if the lag exceeded this range as recommended by Langford et al. (2015).

Since the PTR measures a mixing ratio $\chi_i$ in wet air, some corrections arise in Eq. (5), to account for the dilution of $\chi_{i,d}$ by water vapour. Moreover, since the mixing ratio is computed by normalizing with $\text{cps}^{\text{trans}}_{H_3O^+} + \text{cps}^{\text{trans}}_{H_2O \cdot H_3O^+}$ (Eqs. 1 and 2), the question arises as whether this normalization should be done on raw signal (at 10 Hz) or on averaged signals at 5 min. These two aspects were evaluated by expressing $\chi_{i,d}$ as a function of $\chi_i$ and $\text{cps}_i$ and differentiating Eqs. (1), (2) and (5) (Supplement Sect. 2). The full expression was derived as a function of water vapour flux and $H_3O^+$ covariance (Eq. S11). These corrections remained small in our study: lower than 2 % each for 75 % of the time. Only for a few compounds, the corrections attained 10 % for 15 % of the time. These included noticeably acetone. We finally derived the eddy covariance fluxes, by neglecting the water vapour dilution terms and normalizing by $H_3O^+$ after covariance computation, using the following formula:

$$F_i = S_i \cdot \frac{p_a}{RT_a} \cdot \frac{1.657\,e^{-11}\,U_{\text{drift}}\,T^2_{\text{drift}}}{k\,p^2_{\text{drift}}}$$
$$\cdot \frac{\text{TR}_{H_3O^+}}{\text{TR}_{R_iH^+}\left(\text{cps}^{\text{trans}}_{H_3O^+} + \text{cps}^{\text{trans}}_{H_2O \cdot H_3O^+}\right)} \cdot \overline{w'\text{cps}'_i}.$$ (6)

Note that the relative uncertainty in the flux is higher than that in the mixing ratio, since it includes both the uncertainty in $S_i$ (see above) and the uncertainty in $\overline{w'\text{cps}'_i}$, which

includes high-frequency losses, and uncertainty of the lag. Overall, the flux was calculated over 85 % of the experimental period.

### 2.2.7 High-frequency losses

The magnitude of high-frequency losses was evaluated as the difference between the cross-spectrum of the first water cluster and air temperature in the high-frequency domain, based on the methodology of Ammann et al. (2006). High-frequency losses could not be computed for VOCs due to too high a noise-to-signal ratio, which made the high-frequency part of the spectrum non-exploitable for computing the high-frequency losses. The loss of signal was starting at a frequency around 0.2 Hz, and the signal was halved at around 2 Hz (Fig. S5). Since the power-spectral frequency was at around 0.2 Hz at around 14:00 h UTC, the high-frequency loss appears just after the spectral peak during the day, but the decrease was gentle until 2 Hz and most of the signal energy was contained below 2 Hz. The high-frequency losses were evaluated by comparing the integrated co-spectra (co-ogives) for water vapour cluster and air temperatures. High-frequency losses were evaluated as being less than 5 %. See Ammann et al. (2006) for details. They were therefore not corrected in the following.

### 2.2.8 Fluxes' limit of detection and VOC flux selection

The limit of detection for fluxes (LOD$_f$) was determined as the random uncertainty of the eddy covariance method for each compound, which was calculated as the standard deviation of the ($w'$, $c'$) covariance function at $+80$ and $-80$ s as described by Spirig et al. (2005). We chose 80 s since our base measurement period was 5 min, as opposed to Spirig et al. (2005), who used 180 s with a base measurement period of 30 min. This choice should make no difference, since the turbulence decorrelation time at the measurement height is much shorter than 80 s. The VOCs selected for further analysis were those showing an average flux larger than 3 times the averaged LOD$_f$ over the entire duration of the experiment. Note that the average LOD$_f$ was calculated as the square root of the sum of the squared individual LOD$_f$ divided by the number of records, to be representative of the error in the mean, as detailed in Langford et al. (2015).

### 2.2.9 Identification of VOC, fragments, clusters and isotopes

Since the PTR-MS only measured a mass-to-charge ratio $m/z$, VOCs were tentatively identified by comparing their ion masses with literature values (Yáñez-Serrano et al., 2021). Tentative identifications are summarized in Table S3. By analysing correlations between ion peaks, we identified possible fragments and clusters belonging to the same compound. Ions having a Spearman correlation coefficient larger

than 0.99 were considered to stem from the same compound. Correlated ions were mostly isotopes or VOCs having $m/z$ peaks too close to each other to be separated by the instrument resolution but also compounds with a mass difference between 12 and 42 (Table S4a, S4b and S4c). Therefore, the formaldehyde fluxes ($m/z$ 31.018) reported in the present study should be considered with caution due to uncertainties in its dependence on air humidity.

## 2.3 Meteorological data, CO$_2$ and energy fluxes from the ICOS FR-Gri station

Meteorological measurements included wind speed, air and soil temperatures, and humidity as well as rainfall and global, net and photosynthetic active radiation. The vertical profile of air temperature and wind speed was measured with five two-dimensional ultrasonic anemometers (Wind Sonic, GILL, UK) and shielded thermocouples (HMP155, Vaisala, Finland) placed at 0.5, 1.0, 2.0, 3.0 and 5.0 m above the ground. The CO$_2$ and H$_2$O, sensible ($H$) and latent heat (LE), fluxes were measured by eddy covariance with a closed-path CO$_2$ / H$_2$O IRGA analyser (Li-7200, LI-COR, USA), which was placed at 0.2 m lateral distance from an ultrasonic anemometer (HS-50, Gill, UK). The high-frequency signals were recorded at 20 Hz by a homemade Labview® program, and the fluxes were computed with EddyPro (version 6.2, LI-COR, USA). All flux measurements were made on a mast at 2.7 m height near the centre of the field (Fig. 1) and averaged, and standard deviations were reported over 30 min intervals.

## 2.4 CH$_4$ and H$_2$O mixing ratios and fluxes

CH$_4$ and H$_2$O mixing ratios were monitored using a quantum cascade laser (QCL, CW-QC-TILDAS76-CS Model, Aerodyne Inc., USA) at 10 Hz. The analyser precision at 1 s was 2 ppb for CH$_4$. The analyser was sampling directly in the eddy covariance line at a flow rate of 7 NL min$^{-1}$ and a pressure of 230 mbar below atmospheric pressure. The lag time varied between 2.0 and 2.3 s for the two compounds similar to what was obtained for the PTR-Qi-TOF-MS. The QCL was calibrated prior to the experiment.

## 2.5 Footprint model and contribution of local flux to mixing ratios

The FIDES concentration and flux footprint model (Flux Interpretation by Dispersion and ExchangeS) was used to evaluate the flux footprint and to infer the contribution of the VOC fluxes to the VOC concentrations measured at the site (Loubet et al., 2010; Carozzi et al., 2013; Loubet et al., 2018). The flux footprint $\varphi(x, y)$ of the measurement mast is the probability density that the measured flux $F$ originates from the field point of coordinates $(x, y)$. The measured flux is then $F = \int \varphi(x, y) f_0(x, y) \, dx \, dy$, where $f_0(x, y)$ is the sur-

face flux at coordinates $(x, y)$. The flux footprint should be distinguished from concentration footprint $h(x, y)$ that relies on the concentration measured at the mast $C$ minus the background concentration $C_{\text{bgd}}$ to the surface flux: $C - C_{\text{bgd}} = \int h(x, y) f_0(x, y) \, dx \, dy$. Assuming $f_0(x, y)$ is constant over the studied field, which is a reasonable assumption for a crop, the previous equations can be integrated to provide integrated footprints:

$$F = f_0 \times \int_{\text{field}} \varphi(x, y) \, dx \, dy = f_0 \times \Phi_{\text{field}}, \tag{7}$$

$$C - C_{\text{bgd}} = f_0 \times \int_{\text{field}} h(x, y) \, dx \, dy = f_0 \times H_{\text{field}}. \tag{8}$$

Here $\Phi_{\text{field}}$ has no units while $H_{\text{field}}$ has units of transfer resistance (s m$^{-1}$). The model is based on the same analytical solution of the advection–diffusion equation as the Korman–Meixner model (Kormann and Meixner, 2001), All details of the model can be found in Loubet et al. (2018).

## 2.6 Dataset and statistical analysis

All data were merged into a common dataset according to their date and time and averaged at hourly time steps. The dataset is available as an R format file in the Supplement. All statistics were performed under R (R version 4.0.1, 6 June 2020).

# 3 Results

## 3.1 Meteorological conditions, crop development, CO$_2$ and energy fluxes

The experiment started after wheat anthesis, during the grain-filling period (Ripening stage, Fig. 3). During that period, the grain was progressively filled up to reach a biomass of around 500 g DW m$^{-2}$, similar to the stems biomass (Fig. 3). Senescence also started at the beginning of that period visible by an increase in the yellow-to-green leaves biomass ratio. The crop height was around 0.8–1.0 m.

The campaign started a week after a major flooding event in the Parisian area characterized by a 3-week rain period that might have affected the crop functioning. However, the CO$_2$ flux was in the range of expected fluxes for a wheat crop at that period. The main wind direction was west–southwest, the average wind speed 1.8 m s$^{-1}$ with a typical daily pattern showing night-time low wind speeds except for a few short windy periods (11–13, 20–21 and 28–31 June), accompanied by lower temperatures, higher relative humidity and consistent precipitations (Fig. S6). Air temperature was 17.7 °C on average and varied from 8 to 31 °C with hot periods around 27 June and 7 and 17 July. The period from 12 to 23 June was quite wet, while the driest periods lasted from 7 to 12 June and from 4 to 11 July. The albedo decreased slightly from 0.23 at the start of the experiment (around anthesis) to 0.18 on 23 June (grain-filling period) and remained

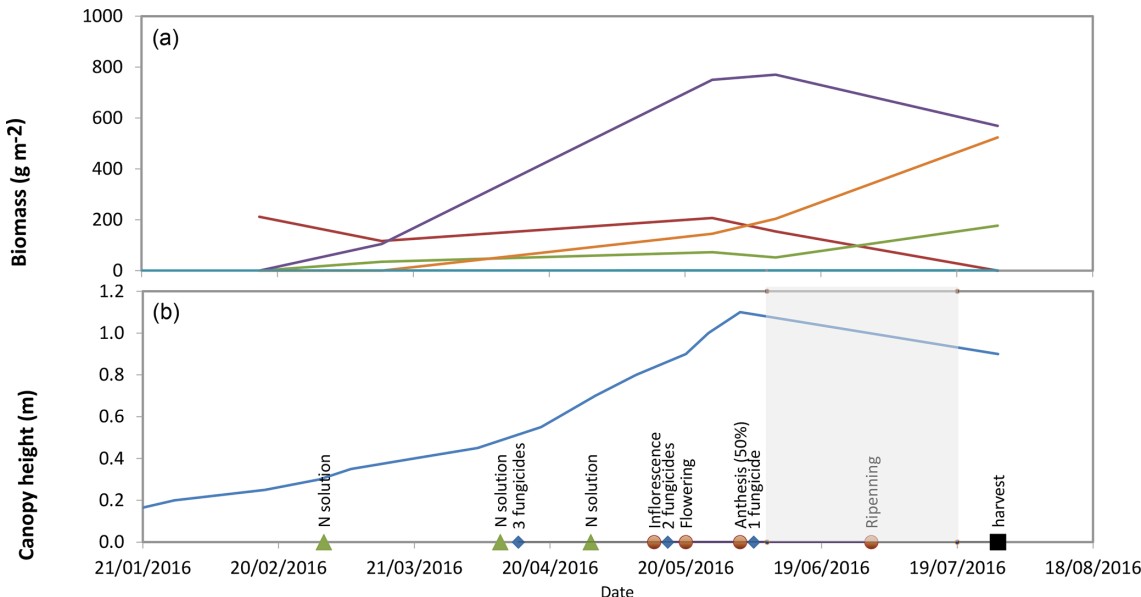

**Figure 3. (a)** Evolution of the aboveground biomass of different plant compartments. **(b)** Crop height, crop developmental stages and farmer activity. The experimental period is highlighted in grey shading.

stable afterward. The air temperature at the crop roughness height $z_0$ ($T_{z0}$), which is a proxy for the plant surface temperature, varied from 10 to 45 °C. The highest temperatures were observed towards the end of the campaign when at midday $T_{z0}$ exceeded the air temperature by up to 10 °C. Before that period, the air-to-canopy temperature difference did not exceed 5 °C at midday. The soil temperature (measured at 5 cm depth) followed the air temperature pattern but with daytime maxima lower than air temperature by 5 to 15°C and night temperature slightly higher than air temperatures due to the canopy shading. The heatwave at the end of the campaign corresponded to very high water vapour pressure deficits at $T_{z0}$ (larger than 4 kPa), while it was usually smaller than 2 kPa before. Night periods usually exhibited dew formation except towards the end of the campaign.

The typical friction velocity of the site was around $0.3 \, \text{m s}^{-1}$, although it was quite high during the windy periods (second to the fourth week), reaching $0.5 \, \text{m s}^{-1}$ (Fig. 4). The exchange velocity for water vapour $V_{max}$ ($H_2O$), which is the physical limitation for convective exchanges, varied daily and showed maximum exchange rates of $3 \, \text{cm s}^{-1}$ at midday and minimum values at night near zero except during the windy periods. The $CO_2$ fluxes showed net absorption during daytime at the beginning of the experiment. They were fairly constant until 15 June and then began to decline until 5 July (first senescence phase), when they switched to net daytime respiration (second senescence phase). Water vapour fluxes decreased less during the same period and showed a high correlation with net radiation but less with vapour pressure deficit. This indicated that, although the canopy was absorbing less $CO_2$, it was still transpiring, and

hence the stomata were open. After week 3, the water vapour flux was rather small which indicates that at that time the crop was fully senescent and stomata were not responding to light anymore. Because of the decrease in evaporation and under quite large net radiations, the sensible heat flux increased from the 20 June onward, while the ground heat flux daily variations sharply increased after the 5 July. The flux footprint from the main field was mostly above 0.8 (median 0.86, interquartile 0.76–0.91) but showed some consistent periods with a lower footprint (down to 0.4) when the wind was blowing from the south. The periods with a footprint lower than 0.6 occupied 13 % of the time.

## 3.2 VOC mixing ratios

The major VOCs at the site were methanol, formaldehyde, ethanol, furan, acetic acid, acetone, hydroxyacetone, acetaldehyde, isoprene and monoterpenes (Table S3). The mixing ratios of the most emitted and deposited VOCs showed no marked daily patterns but some similarities in their weekly patterns (Fig. S7). During the first week, mixing ratios were lower and diurnal variations smaller than during the rest of the experimental period. This weak corresponded to a rainy period with westerly winds typical of oceanic influence at the site (Fig. S6). The second week showed an increase in the mixing ratios of all compounds that mostly lasted for 3 weeks. This period corresponded to the end of the rainy period, a sudden change in the wind direction to the east and an increase in temperature and water vapour content of the atmosphere (Fig. S6). This event also brought polluted air masses from the Parisian area as shown by the increase in

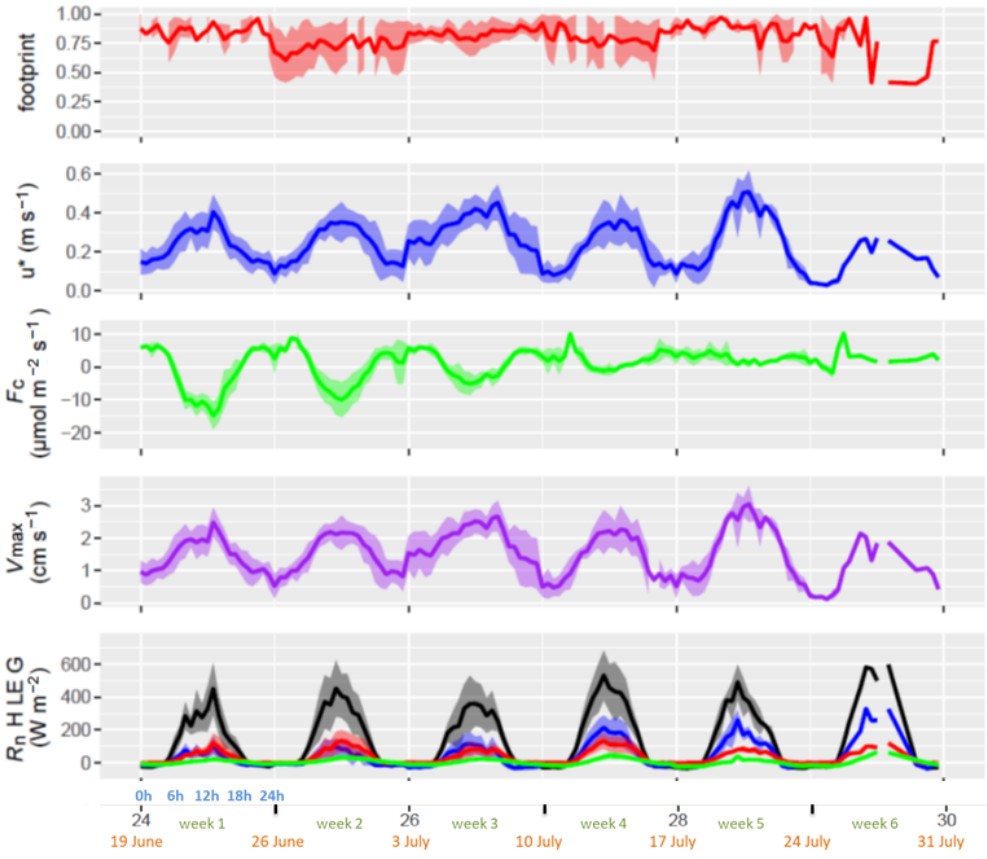

**Figure 4.** Field flux footprint, friction velocity ($u_*$), $CO_2$ flux ($F_c$), maximum exchange velocity for water ($V_{max}$) and terms of heat balance ($R_n$ $H$ LE $G$): the net radiation ($R_n$, black), sensible heat flux ($H$, blue), latent heat flux (LE, red) and ground heat flux ($G$, green). By convention, except for $R_n$ and $G$, positive fluxes are ecosystem losses and negative fluxes are gains or uptakes. Each week shows the mean of diel cycles (lines) plus standard deviations (ribbons). The $x$ axis shows the week number in the year (black), the week number within the experimental period (green), the starting date of the week (orange) and the hour of day (blue).

$NO_2$ (Fig. S7). Most compounds showed a sharp increase following rain stop and a drop after 36 weeks.

This period also showed a concentration increase in several oxygenated, mainly deposited compounds like hydroxyacetone, 4-oxopentanal, methylfuran or propyne (Fig. S7). Week 3 corresponded to the windy period (Fig. 4) with a well-mixed boundary layer both during day and night and to wind blowing from the farm. Week 4 is marked by maximum concentrations of methanol and acetaldehyde but also of most of the oxygenated compounds, the daily pattern of these being more marked during this week. This period, which corresponded to the end of the senescence period, was also characterized by high air temperatures (up to 30 °C) and high surface temperatures (up to 40 °C) (Fig. S6) as well as by peak $O_3$ and NO mixing ratios, due to the first weekend of the summer holidays in France and a huge traffic rush in the region (Fig. S6). Finally, the last week corresponded to a very warm period with low wind speeds.

## 3.3 VOC fluxes

In total, 123 VOCs had fluxes greater than 3 times the flux detection limit ($LOD_f$) when computed over the whole period. However, when expressed over hourly periods, only four VOCs showed an average flux larger than 3 times the mean $LOD_f$. This can be attributed to the fact that the $LOD_f$ diminishes over long integration periods because it is essentially a random error. Of these 123 VOCs, 32 were on average emitted, while 91 were on average deposited.

The most emitted compound was methanol ($m/z$ 33.033) with an average molar flux rate of 0.5 nmol m$^{-2}$ s$^{-1}$, which was, respectively, 2 and 6 times higher than the second and third most emitted VOCs ethanol ($m/z$ 47.049) and furan ($m/z$ 93.033) (Fig. 5, Table S3). Acetaldehyde ($m/z$ 45.0433) was the fourth most emitted compound, acetone ($m/z$ 59.049) the fifth and dimethyl sulfide (DMS, $m/z$ 63.026) the sixth. Methanol emissions increased during the first month, decreased temporally during the fifth week to increase again during the last week of July. The daily pattern of methanol fluxes showed a sharp increase in the late

morning followed by a maximum at around 12–15 h and a minimum at the end of the night. Ethanol flux increased only after week 5. Furan ($m/z$ 93.033) showed a different seasonal pattern with the largest emissions during the second week and then decreasing emissions towards the end of July, suggesting less dependency to temperature. Acetaldehyde fluxes showed a similar trend as methanol, suggesting that both are linked to a similar process. Acetone fluxes showed a rather stable daily pattern during the period, except for a slightly larger flux during week 4. DMS fluxes showed a positive peak at the end of the rain period (middle of the second week) and a decrease afterward. In addition, DMS fluxes slightly increased during the hottest period (senescence).

The most net-deposited VOCs were formaldehyde ($m/z$ 31.018), formic acid ($m/z$ 47.013), ion $m/z$ 43.018 (which comprises fragments of several oxygenated compounds), hydroxyacetone ($m/z$ 75.044), acetic acid ($m/z$ 61.028), methylfuran and 2-methyl-3-buten-2-ol (MBO) ($m/z$ 83.049) (Fig. 6). These compounds contributed to 29 %, 15 %, 11 %, 9 %, 6 % and 3 % of the deposition fluxes summed over all compounds that passed the $\mathrm{LOD_f}$ criteria, on a molar basis, respectively. The VOC deposited showed distinct patterns: formaldehyde ($m/z$ 31.018) showed a consistently larger deposition rate during the first 2 weeks, followed by a decrease in deposition in the following weeks, switching to a net emission during the fifth week. Formic acid ($m/z$ 47.013) and methylfuran/MBO ($m/z$ 83.049) showed a small deposition during the first 2 weeks followed by an increase in week 3, a maximum in week 4 and a decrease during week 5. Hydroxyacetone ($m/z$ 75.044) was deposited consistently throughout the period with larger deposition fluxes during weeks 2 to 4, mirroring the emission behaviour of furan ($m/z$ 93.033). Ion $m/z$ 43.018, corresponding to fragments of several ions, and acetic acid ($m/z$ 61.028) showed a similar pattern of bidirectional fluxes with small deposition or emission rates during the first 2 weeks, followed by large deposition fluxes during week 3 that decreased during the last weeks ending in small emissions.

### 3.4    Total VOC emission and deposition fluxes

The sum of emissions of all VOCs was $1.02 \pm 0.04\ \mathrm{nmol\ m^{-2}\ s^{-1}}$ (mean ± the sum of all $\mathrm{LOD_f}$), which corresponds to $41\ \mathrm{g\ ha^{-1}\ d^{-1}}$ ($\sim 20\ \mathrm{g\ C\ ha^{-1}\ d^{-1}}$). On a molar basis, methanol contributed to 52 % of the total VOC emissions, ethanol 23 %, furan 9 %, acetaldehyde 6 %, acetone 4 % and DMS 1.3 %. The average total deposition rates amounted $0.5 \pm 0.01\ \mathrm{nmol\ m^{-2}\ s^{-1}}$, which corresponded to $26\ \mathrm{g\ ha^{-1}\ d^{1}}$ ($\sim 12\ \mathrm{g\ C\ ha^{-1}\ d^{1}}$). Interestingly, the total VOC deposition rate was 52 % of the total emission rate on a molar basis but was around 63 % when expressed on a mass basis. This shows that the deposited compounds were on average heavier than the emitted ones. Additionally, the daily pattern of deposition fluxes showed two peaks, of

which the later one coincided with the afternoon rush hour. This suggests that a great fraction of deposited VOCs stems from traffic. Indeed, parallel NO and $NO_2$ measurements made by Vuolo et al. (2017) showed that the site was under the advection of the roads on the north, west and east and from the Parisian area. The first peak was around noon, suggesting a biological or chemical source (Fig. 7). Overall, the summed emission and deposition fluxes led to a net flux (emission) of $\sim 15 \pm 2\ \mathrm{g\ ha^{-1}\ d^{-1}}$ ($0.5 \pm 0.05\ \mathrm{nmol\ m^{-2}\ s^{1}}$), which approximates to $\sim 8 \pm 1\ \mathrm{g\ C\ ha^{-1}\ d^{-1}}$.

## 4    Discussion

### 4.1    VOC flux eddy covariance measurements with a PTR-Qi-TOF-MS: pitfalls and open questions

Because of the air density fluctuations, the Webb–Pearman–Leuning (WPL) correction needs to be applied to the eddy covariance flux (Webb et al., 1980). This correction accounts for dilution by water vapour and temperature-induced density variations in the analysed air. Because of the long and heated sampling lines and large thermal mass of the drift tube, temperature variations can be neglected. Water vapour dilution however needs to be accounted for, since the water vapour fluctuations are not directly measured in the PTR-MS. Eq. (S9) shows the WPL correction specific to the PTR-Qi-TOF-MS. We found that most of the time this correction was smaller than a few percent (Fig. S1) because of the relatively small mixing ratios of the measured VOCs in ambient air.

Another issue that, to our knowledge, was not discussed before concerns the effect of normalizing the cps by the cps of $H_3O^+$ on 10 Hz data prior to calculating the eddy covariance fluxes. We showed that the bias is a function of the $\overline{w'\mathrm{cps}_{H_3O'^+}}$ covariance, which is not null since $H_3O^+$ is consumed by VOCs and water vapour that are themselves correlated to vertical wind speed (Eqs. S13 and S14). However, this bias is small and negligible when integrated over time (Figs. S2 and S3). Nevertheless, we recommend calculating the covariances using raw cps and normalizing them by the primary ion $H_3O^+$ afterwards, to avoid this minimal though proven bias. This is important especially in conditions with very strong water vapour or VOC fluxes since $\overline{w'\mathrm{cps}_{H_3O'^+}}$ may increase under such conditions.

High-frequency losses were evaluated to be small and in line with the expected values for the lag time of our sampling system (lower than 5 %) (Ammann et al., 2006). These high-frequency losses were estimated from the first water cluster ($m/z$ 37.028, Fig. S5). Based on the comparison of the shapes of the cross-correlation functions for the water cluster and methanol (Fig. S4), we hypothesize that the high-frequency losses for methanol should be similar to that for water vapour. Since methanol is a sticky compound, we expect that this would be also true for most measured VOC. However, we were unable to evaluate high-frequency losses

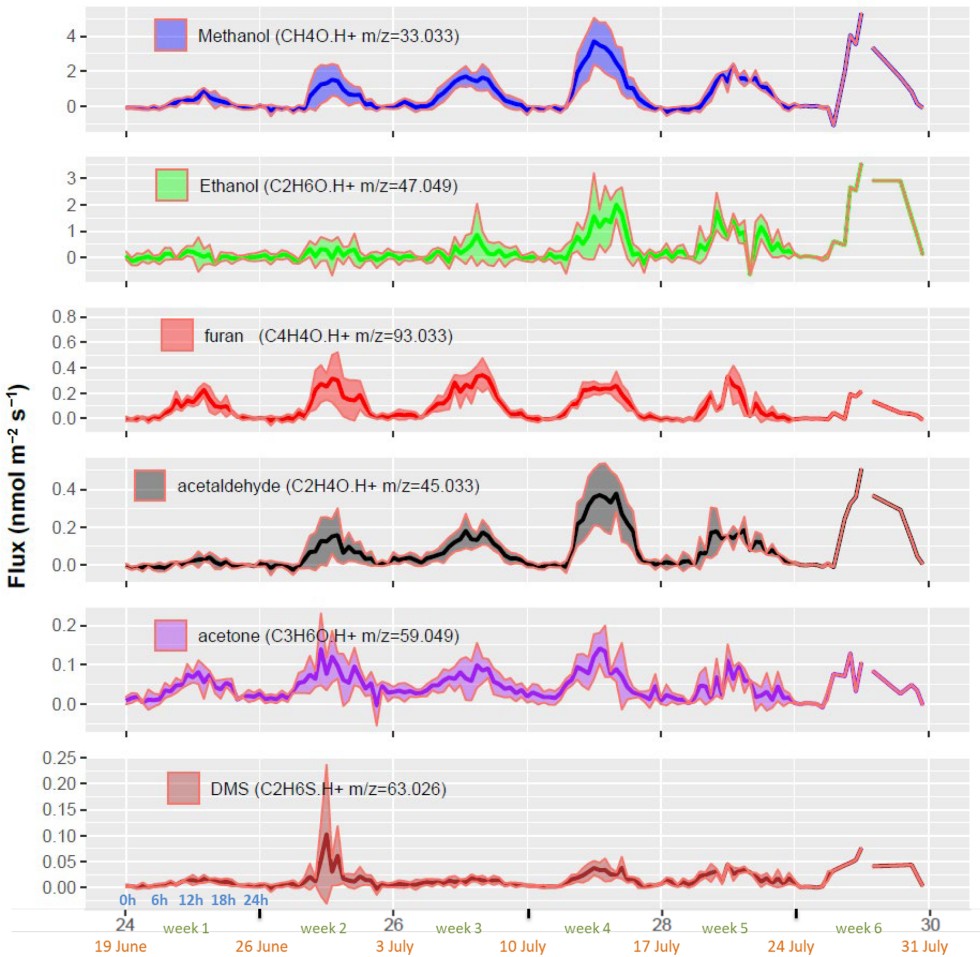

**Figure 5.** Whole ecosystem net fluxes of the six most emitted VOCs. Each week shows the diel cycle with its mean (line) and standard deviation (ribbons). The x axis denotes the week number in the year (black), the week number in the experiment (green), the starting date of the week (orange) and the hour of day (blue). Note that no standard deviations are shown for week 6 because only 1 d of measurement was available.

for methanol based on its w.cps CE1 cross-spectra, which showed higher values at high frequencies than at lower ones (Fig. S5). We interpret this as an effect of the noise-to-signal ratio of the VOC cps (Langford et al., 2015). Although we expected some large noise contribution to the variance of the VOC signals, we did not expect a contribution to the w.cps cross-spectra. The cross-spectrum exemplified in Fig. S5 however shows the opposite. A reconstruction of the noisy signal mimicking that of methanol (not shown) confirms that noisy signals are difficult to analyse for high-frequency losses since the high-frequency end of the signal is dominated by the noise. To tackle this issue, one would need to develop some denoising algorithm, which is beyond the scope of this paper.

## 4.2   Magnitude of major VOCs exchanged

Overall, we were able to identify more than 123 VOCs that had fluxes larger than 3 times the flux limit of detection ($LOD_f$). Park et al. (2013a) reported almost 500 VOCs whose fluxes exceeded 3 times the $LOD_f$. It is not fully surprising that we found less VOCs showing a significant flux, since Park et al. (2013a) measured above an orange orchard in the Central Valley of California in the summer, which is a much more emitting ecosystem in a period favourable to emissions (Park et al., 2013b). Indeed, they found a methanol flux ($73\,\mu g\,C\,m^{-2}\,h^{-1}$) around 3 times larger than in this study ($23\,\mu g\,C\,m^{-2}\,h^{-1}$) and similarly for acetone and acetaldehyde. Overall, their fluxes were hence at least 3 times larger than ours, which probably explains the larger number of compounds having a flux above the $LOD_f$ in their study. Interestingly however we found around the same number of deposited compounds ($\sim 190$) (Park et al., 2013a). Note that, although the most deposited compound was formaldehyde, it

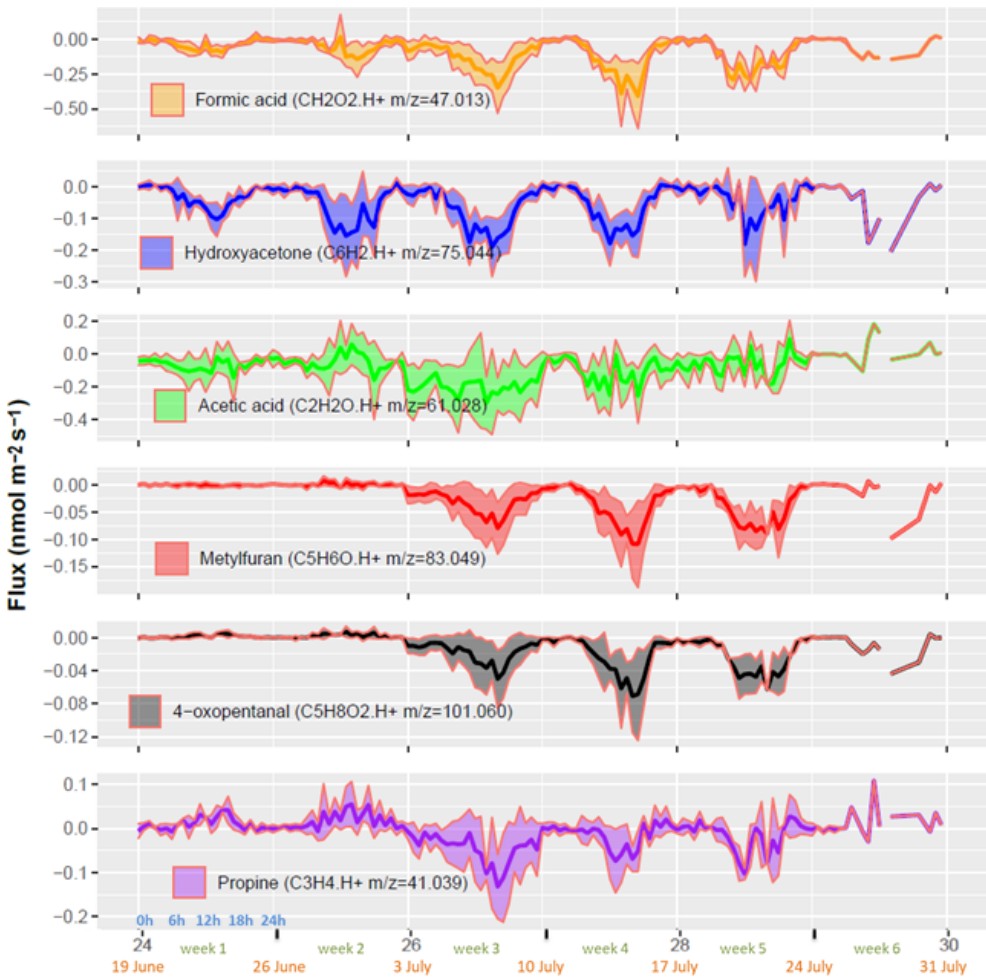

**Figure 6.** Whole ecosystem fluxes of the six most deposited VOCs. Each week shows the diel cycle with its mean (line) and standard deviation (ribbons). The $x$ axis shows the week number in the year (black), the week number in the experiment (green), the starting date of the week (orange) and the hour of day (blue). Only 1 d of measurement was available during week 6, leading to no interquartile computation and hence no ribbons.

is not discussed here since the uncertainties in its concentration measurement, especially in response to water vapour, are too large in this study to provide consistent measurements.

In our study methanol ($m/z$ 33.033) was by far the most emitted VOC. Jacob et al. (2005) estimated that growing vegetation is the major source of methanol at the global scale (128 Tg yr$^{-1}$) followed by a recombination of methylperoxyradicals ($CH_3O_2$) with itself and other organic peroxy radicals (38 Tg yr$^{-1}$), plant decay (23 Tg yr$^{-1}$), biomass burning and biofuels (13 Tg yr$^{-1}$), and vehicles and industry (4 Tg yr$^{-1}$). The very few studies on VOC fluxes from wheat (Table 1) reported that methanol is the most emitted compound, representing 20 % to 80 % of the overall VOC fluxes (Bachy et al., 2020; Gonzaga Gomez et al., 2019). Hence, our results corroborate these previous studies suggesting that cereal crops represent a potentially strong source of methanol, at least for winter wheat in Europe.

More specifically, our study confirms the increase in methanol emissions during senescence (Fig. 5) as already observed by Bachy et al. (2020) and Gonzaga Gomez et al. (2021) for wheat and Mozaffar et al. (2018) for maize. These observations collectively suggest that cereal crops become a major source of methanol at the end of the cultural cycle with net emission rates possibly exceeding those during the vegetative growth phase. This seasonal peak can be explained by the demethylation of pectin in senescent plant tissues (Fall and Benson, 1996) along with the pronounced degradation of cellular components at the end of chlorosis (Keskitalo et al., 2005; Woo et al., 2019), both of which promote methanol production and its release through higher leaf porosity. In addition, methanol emissions were enhanced by high tissue temperatures during the senescence period due to warm weather conditions and reduced transpiration (Figs. S6 and 6).

Atmos. Chem. Phys., 22, 1–26, 2022                                   https://doi.org/10.5194/acp-22-1-2022

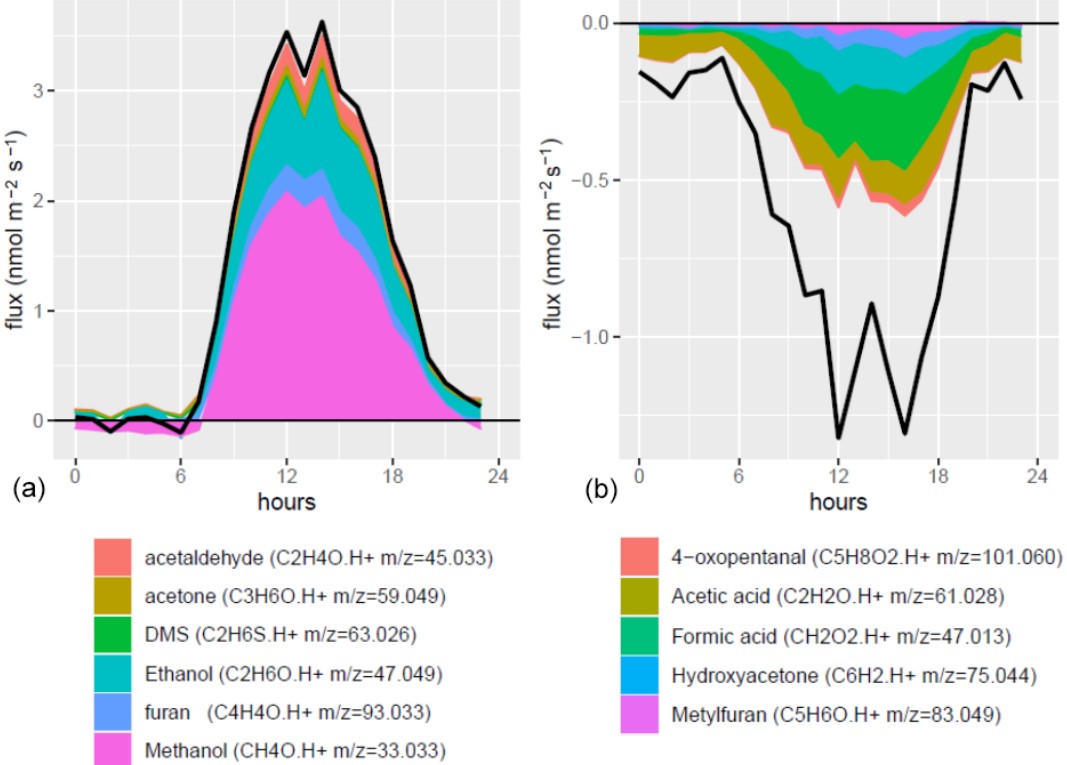

**Figure 7.** Stacked daily averages of the six most emitted **(a)** and six most deposited **(b)** VOCs. Black lines show summed-up emissions and depositions for compounds with fluxes larger than $LOD_f$.

We recorded rather small deposition rates at the end of the night, when the canopy was the wettest (Fig. 5). As a very soluble compound, methanol can be easily adsorbed in water layers and desorbed when the canopy dries out (Laffineur et al., 2012; Bachy et al., 2018). However, we did not find any clear desorption pattern in the morning, suggesting instead moderate net surface deposition at night in our study.

Ethanol ($C_2H_6O$, $m/z$ 47.028) is an intermediate in the production of acetaldehyde under anoxic conditions in the root system by fermentation. After transport to the leaf level, ethanol is oxidized to acetaldehyde, which can be further oxidized into acetate (Kreuzwieser et al., 2001; Niinemets et al., 2014; Seco et al., 2007). Indeed in our study, ethanol and acetaldehyde emissions expressed a similar pattern suggesting that the two compounds originate from closely associated processes, which is in line with the MEGAN model, which assigns the same emission factor to ethanol and acetaldehyde (Guenther et al., 2012). Ethanol was also identified as a dominant VOC emitted by corn silage due to fermentation, typically making up 3 % of the US VOC emissions (Hafner et al., 2010; Montes et al., 2010).

Furan emissions from crops (tentative identification of $m/z$ 93.033) have not been reported in previous studies so far, apart from the work of Gonzaga et al. (2019) using plant chambers with the same instrument at the same site (Table 1). In their study they observed maximum fluxes during the grain-filling period (Fig. 4). To our knowledge, the only other studies reporting emissions of this compound are those by Stockwell et al. (2016) from biomass burning, Venneman et al. (2020) from root endophytic fungi (*Serendipita*) and Park et al. (2013b) from an orange orchard. Park et al. (2013b) reported an average flux of $4.2\,\mu g\,C\,m^{-2}\,h^{-1}$ over 24 h, which is about 5 times lower than what we observed here, suggesting that this compound may be specific to wheat (Table 1). Other studies mentioned furan but either identified it as a compound emitted from the chamber material or a compound the flux of which remained below the detection limit (Batten et al., 1995; Gitelson et al., 2003). In our study, because of the method used to measure the flux, a desorption process from the tubing is not plausible. Indeed, this would mean that desorption would be correlated with vertical wind speed at high frequency, which is not likely.

Acetaldehyde ($m/z$ 45.033) was the third most emitted compound with a flux range similar to that reported in previous studies on wheat (Table 1). The measurements from Gonzaga et al. (2019) with dynamic chambers show remarkable agreement during the same grain-filling period, suggesting the plant is the main source at that time. Bachy et al. (2020) determined significantly lower average values, which is explained by the fact that their values refer to the entire season, whereas our values refer to the grain-filling period. The emission pattern of acetaldehyde was very similar to that of

methanol, suggesting a similar production process leading to emissions or at least a similar dependency of these processes on environmental conditions and plant and soil physiology (Figs. 4 and 5). However, the mechanisms leading to acetaldehyde emissions are not well characterized yet, especially in the field. Enhanced acetaldehyde emissions were observed following leaf wounding (De Gouw et al., 2000; Graus et al., 2013), light–dark transition (Karl et al., 2002), and exposure to oxidative (ozone) and anoxic stresses (Seco et al., 2007). The latest observation may be particularly relevant for the present study, since the wheat crops at our site are frequently exposed to high ozone deposition fluxes and thus oxidative stress (Tuzet et al., 2011; Stella et al., 2013; Potier et al., 2015; Vlasenko et al., 2010). Potier et al. (2017) have shown that ozone reacts strongly with wheat leaf water extracts and especially when wheat is senescent. Moreover, water treatment ozonation studies have shown that organic matter oxidation is a source of acetaldehyde (Papageorgiou et al., 2014). Together these studies suggest that acetaldehyde emission from crops through the oxidation of organic plant or soil material by ozone may be a significant emission source. This source would be especially significant during senescence when the cells degrade and organic matter is exposed to the atmosphere. This hypothesis is supported by the work by Potier et al. (2015, 2017), showing an increased ozone deposition during senescence over wheat.

Globally, In our study, acetone ($m/z$ 59.049) was among the most emitted compounds, similarly to what Bachy et al. (2020) and Gonzaga et al. (2019) measured over wheat. Acetone is known to be mostly emitted by terrestrial vegetation ($\sim 30\%$ of the total emissions) and oceans ($\sim 30\%$), with additional sources being plant decay (2%–10%) and the oxidation of isoalkanes in the atmosphere (Jacob et al., 2005). Maillard reactions in dead decaying plant material were identified as a potentially important source of acetone (Warneke et al., 1999). Increased acetone emissions from intact plants were observed upon stresses including injury (Davison et al., 2008), ozone exposure and water logging (Cojocariu et al., 2005). Fruekilde et al. (1998) suggested that acetone is produced at the plant surfaces from the ozonolysis of epicuticular waxes. The available literature therefore suggests that acetone emissions by plants may significantly come from non-enzymatic reactions occurring during the degradation of plant cells and surfaces. Soil may be an additional source of acetone, increasing with soil organic matter (Abis et al., 2018; Schade and Goldstein, 2001; Zhao et al., 2016). In particular, acetone was proved to be a secondary product of the cyanogenic pathway (Seco et al., 2007).

To date dimethyl sulfide emissions (DMS, $m/z$ 63.026) from terrestrial plants have been rarely reported. The DMS fluxes observed in our study are in very good agreement with those from Kanda et al. (1995), who measured DMS emissions from maize and wheat with static chambers, a technique which may however be criticized for exposing the plants to unnatural, non-steady-state environmental conditions (Niinemets et al., 2011). Kanda et al. (1995), however, clearly showed that the aboveground part of the wheat was the source. Yonemura et al. (2005) found emissions from *Hibiscus spec.* CE6 in quantities comparable to our studies ($\sim 6\,\mu g\,m^{-2}\,h^{-1}$), whereas Vettikkat et al. (2020) reported large emissions of DMS by tropical forest trees (Mahogany tree, *Swietenia macrophylla*) in the range of 7–40 $\mu g\,m^{-2}\,h^{-1}$ (assuming an LAI of $\sim 7\,m^2\,m^{-2}$ as observed in the present study). Jardine et al. (2015) reported large DMS concentrations in the Amazon forest and showed that a soil source was present. Globally, the largest identified DMS sources so far are oceans ($\sim 100$ times larger than terrestrial sources), through the dimethylsulfoniopropionate (DMSP) production by phytoplankton in their photosynthetic cycle (Groene, 1995). Interestingly, DMSP was also found to be emitted from several terrestrial species and to increase under drought stress (Haworth et al., 2017). In our study we noticeably found larger DMS emissions during the second week (Fig. 5), which corresponded to a period with a consistently larger atmospheric water vapour concentration and some rain. This suggests that DMS emissions from wheat are promoted by wet conditions. However, Gonzaga Gomez et al. (2019) did not find DMS to be emitted by wheat leaves in amounts similar to our study, suggesting an emission from the soil during that period. Carrión (2017) suggested soil DMS emissions from methanethiol degradation by soil bacteria. Indeed, Abis et al. (2018) found emissions from sieved soils, though small in magnitude. Furthermore, Venneman et al. (2020) showed clear emissions of DMS from root endophytic *Serendipita* fungi. Clearly, DMS emissions by terrestrial ecosystems, and soils in particular, need further investigations.

Crops are generally regarded as low emitters of isoprene ($m/z$ 69.070) (Lathière et al., 2010). Indeed in our study, isoprene was mostly deposited, except during the first week (Fig. 5). Bachy et al. (2020) also reported deposition events, especially during senescence in the early morning. Since isoprene is fragmenting and other compounds fragment on $m/z$ 69.070, the observed deposition of $m/z$ 69.070 may include compounds other than isoprene (Zhou et al., 2017; Karl et al., 2012). This issue is discussed in the Supplement (Fig. S11). In particular, assuming that ion $C_5H_8^+$ ($m/z$ 68.050) is a proxy of isoprene, this ion mostly showed emissions except for week 28, suggesting that isoprene may be emitted by wheat, but MBO and other fragmenting compounds would be mostly deposited. Bearing in mind that this is a hypothesis, we conclude that isoprene was mostly emitted at a rate close to Morrison et al. (2016) and similar to Bachy et al. (2020).

Monoterpenes (MT, $m/z$ 137.129) were mostly deposited (Table 2), with a flux range similar to that of Bachy et al. (2020). We found that the cps of ions $m/z$ 81.070 and 137.129 were highly correlated (correlation coefficient = 0.98), with a ratio of 81/137 of 2.5 (Table S4), in agreement with previous studies (Tani et al., 2003; Tani et

**Table 1.** Fluxes and mixing ratios of the 9 CE2 most emitted VOCs found in this study, together with isoprene and monoterpenes, compared to literature values using different methods of measurement. VOC fluxes measured by eddy covariance refer to the whole ecosystem including soil and are expressed per square metre of ground surface. Fluxes from chamber measurements refer to projected surface and dry weight (DW) of the enclosed aboveground organ of wheat. Means $\pm$ standard errors and [min–max] ranges.

| $m/z$ | Tentative identification | Mixing ratio ppb | Flux $\mu g\,m^{-2}\,h^{-1}$ | Flux $ng\,g^{-1}\,DW\,h^{-1}$ | Measurement method | Reference |
|---|---|---|---|---|---|---|
| 33.033 | Methanol | 3.4 | $63 \pm 4$ | $30 \pm 2$ | Eddy cov. | This study |
|  |  |  |  | [680–1100] | Dyn. chamb. | G2019 |
|  |  | [1–10] | $62 \pm 3.3$ | [−255–710] | Eddy cov. | B2020 |
| 47.049 | Ethanol | 1.7 | $41 \pm 4$ | $20 \pm 2$ | Eddy cov. | This study |
| 93.033 | Furan ($C_6H_4O$) | 1.7 | $30 \pm 1.5$ | $15 \pm 1$ | Eddy cov. | This study |
|  |  |  |  | [10–50] | Dyn. chamb. | G2019 |
| 45.033 | Acetaldehyde | 0.3 | $9.6 \pm 0.6$ | $5 \pm 0.4$ | Eddy cov. | This study |
|  |  |  |  | [10–50] | Dyn. chamb. | G2019 |
|  |  |  | $-2 \pm 0.8$ | [−80–75] | Eddy cov. | B2020 |
| 59.049 | Acetone | 0.7 | $9.1 \pm 0.3$ | $4.5 \pm 0.15$ | Eddy cov. | This study |
|  |  |  |  | [80–180] | Dyn. chamb. | G2019 |
|  |  |  | $-2 \pm 0.8$ | [−75–75] | Eddy cov. | B2020 |
| 63.026 | DMS | 0.1 | $2.9 \pm 0.15$ | $1.5 \pm 0.1$ | Eddy cov. | This study |
|  |  |  | [0–11.6] | [0–14.5] | Static chamber | K1995 |
|  |  |  | [0.2–0.5] | 0.03 | Dyn. chamber | F1988 |
| 95.049 | Phenols | 0.1 | $3.2 \pm 0.3$ | $1.6 \pm 0.1$ | Eddy cov. | This study |
| 69.070 | Isoprene + fragments | 0.2 | $-1.3 \pm 0.2$ | $-0.6 \pm 0.1$ | Eddy cov. | This study |
|  |  |  | 4.8 | [0–6000] | Dyn. chamb. | M2016 |
|  |  |  | [−10–25*] | [−5.5–14] | Eddy cov. | B2020 |
|  |  |  |  | [0–50] | Dyn. chamb. | K2009 |
| 137.132 | Monoterpenes | 0.2 | $-2.6 \pm 0.1$ | $-1.3 \pm 0.05$ | Eddy cov. | This study |
|  |  |  |  | [−50–18] | Dyn. chamb. | G2019 |
|  |  |  | [0–12 000] | [0–420 000] | Dyn. chamb. | M2016 |
|  |  |  | [−10–25*] | [−5.5–14] | Eddy cov. | B2020 |

F1988: Fall et al. (1988). K1995: Kanda et al. (1995) CE3 . K1995: Konig et al. (1995). K2009: M. Karl et al. (2009). B2020: Bachy et al. (2020). G2019: Gonzaga Gomez et al. (2019). M2016: Morrison et al. (2016). In this study, 18 T ha$^{-1}$ dry biomass, which is the mature wheat field biomass, was used as a scaling parameter. K1995: closed chamber measurements were performed over 10 min twice a day in PVC chambers. CE4 * Rough estimations based on averaged diurnal cycles. CE5 # $m/z$ 68.06, for which $C_5H_8^+$ is used as proxy. It is multiplied by 12 to 24, which is the slope of $m/z$ 69.07 to $m/z$ 68.06 at $E/N = 150$ and $E/N = 130$, respectively.

al., 2004; Steeghs et al., 2007; Misztal et al., 2012). This ratio and the correlation coefficient were stable whatever the $E/N$, though slightly higher at $E/N = 150$ (2.7). However, interestingly the $m/z$ 137.129 deposition flux was larger than that of $m/z$ 81.070, while the mixing ratios were ranked in an opposite way. This indicates that the monoterpenes were composed of a mix of compounds having different deposition velocities and different fragmentation patterns. Indeed the ratio of $m/z$ 81/137 was shown to depend on the monoterpene compound and $E/N$ (Misztal et al., 2012). According to their study, and bearing in mind the many uncertainties, the measured ratio 81/137 in our study would correspond to 3-carene. Deposition of monoterpenes was observed over grassland by Bamberger et al. (2011) during an episode when the surrounding pine forest was damaged by a hail-

storm, hence showing that monoterpenes can be deposited when ambient concentration is high. In our study, the wheat field was next to a woodland and farm, both being a source of monoterpenes (Kammer et al., 2020) that clearly enhanced the mixing ratios of MT under favourable wind conditions (Fig. S13). This suggests that the MT deposition was governed by local advection from surrounding sources. Deposition of monoterpenes was also demonstrated by modelling studies even for low MT-emitting canopies, confirming a bidirectional behaviour of MT fluxes (Zhou et al., 2017).

Phenol ($m/z$ 95.049) emissions from terrestrial surfaces are mostly due to biomass burning but also to fresh manure and livestock (Kammer et al., 2020). Phenol is common in decomposing organic material. However, we found no literature data on phenol emissions from plants or soils. Our

experimental field receives a lot of manure and slurry (Loubet et al., 2011), which together with the organic matter of senescing plants may have mostly contributed to the overall phenol emissions. Phenolic compounds, especially phenolpropanoids and benzenoids, are a class present in all plants and essentially produced via the shikimate pathways from amino acids (Fares et al., 2010) and noticeably in wheat (Kiraly, 1962).

Formic acid ($CH_2O_2$, $m/z$ 47.013) was the second most deposited VOC, which also had high deposition velocities. Link et al. (2020) showed that it is formed from the oxidation of isoprene degradation products – methacrolein (MACR), isoprene epoxydiol (IEPOX), isoprene hydroxy hydroperoxide (ISOPOOH) – with the OH radical. The daily pattern of $V_{dep}$ showed clear deposition during the day and the smallest deposition rate at the end of the night. Formic acid is also known to be produced by organic matter degradation (Holopainen et al., 2017) and several studies observed clear emissions of formic acid from ecosystems (Nguyen et al., 2015; Rantala et al., 2015). Bidirectional fluxes were indeed reported by Brilli et al. (2016) in a European forest and by Jardine et al. (2011) in a tropical forest, who estimated the compensation point to be around 1.3 ppb. In our study, however, we had smaller atmospheric concentrations suggesting that the compensation point was much smaller (lower than 70 ppt).

The third most deposited ion was $C_2H_2O$ ($m/z$ 43.018, Table 2). Although the raw formula of this ion is quite clear ($C_2H_2O \cdot H^+$ TS4) and would correspond to ethenone, it is most certainly a mix of several fragments (2,3 butanedione, acrylic acid, methyl vinyl ketone, acetic acid, formic acid), as shown by the synthesis of Yáñez-Serrano (2021). In particular, it was shown to be a fragment of acetic acid (Baasandorj et al., 2015). We did indeed find a correlation of 0.9 between $m/z$ 61.028 and $m/z$ 43.018 mixing ratios and also observed correlations between the fluxes of the two compounds (Fig. 6). We also found a correlation of 0.84 between $m/z$ 71.049 (methyl vinyl ketone or MVK and MACR) and $m/z$ 43.018 cps. Ion $m/z$ 43.018 deposition velocity did not show a very clear diurnal pattern. However, during the stormy week (week 3), the deposition flux and velocity increased in magnitude (Figs. 6 and 8). The absence of a clear daily pattern in deposition velocity is surprising since $V_{max}$ itself shows a diel pattern. This may be related to a surface resistance that compensates the daily variations in $V_{max}$, namely higher surface resistance during the day and lower ones at night due to the affinity with water of these compounds that are mostly polar. Indeed, at night leaf surfaces were often wet while they were dry during the day, as shown by the wetness index (Fig. S6). Ion $m/z$ 43.018 was also emitted during a short dry period at the start of the senescence, but no emissions or strong deposition were observed when the canopy was either growing of completely senescent. This bidirectional behaviour may be due to the fact that this ion is a combination of several compounds.

Hydroxyacetone was the fourth most deposited compound ($m/z$ 75.044, $C_3H_6O_2$, Table 2). Hydroxyacetone (HAC) is an oxidation product of isoprene after oxidation by $O_3$, OH or NO. High deposition velocities above a forest averaging $1.4\,\mathrm{cm\,s^{-1}}$ have been reported by Nguyen et al. (2015). In our study the deposition velocity of HAC was about 20 times lower, which can be partly explained by the lower turbulence level above a wheat crop with respect to a forest canopy, leading to higher transfer resistances (Seinfeld and Pandis, 1998). We estimate that in our study the maximum exchange velocity ($V_{max}$), which expresses the maximum deposition velocity for a non-reactive compound, was 3 to 4 times lower than in Nguyen et al. (2015). We are not aware of other studies reporting hydroxyacetone deposition over crops. In terms of carbon, HAC deposition represented about one-third of the methanol emission and was 3 times the isoprene emissions (Table 1). Regarding its diurnal variation, the deposition velocity showed a very clear pattern with an increase early in the morning, quite a constant $V_d$ throughout the day and a decrease late in the evening (Fig. 8). Nguyen et al. (2015) found a similar increase in the morning but a decrease that started earlier during the day and became enhanced towards the evening. The difference between these studies may suggest a difference in the non-stomatal deposition or in the daily turbulence pattern between studies. Products of atmospheric VOC oxidation like HAC can be taken up by plants and metabolized as shown for example by Karl et al. (2010).

The fifth most deposited compound was acetic acid ($m/z$ 61.028). Acetic acid is formed in the atmosphere by the oxidation of isoprene and its oxidation products, with $O_3$ and OH at low NO (Link et al., 2020). Acetic acid can also be directly emitted from soils (Mielnik et al., 2018) and from plants especially under stress conditions and during senescence (Portillo-Estrada et al., 2020). Kesselmeier et al. (1998) found emissions of acetic acid from trees but only deposition on barley and other crops, suggesting a bidirectional stomatal uptake and subsequent usage by plant metabolism, confirmed by Staudt et al. (2000). In the present study, acetic acid deposition velocity did not show a clear diurnal pattern though $V_d$ was slightly larger in the morning than in the afternoon (Fig. 8). This behaviour may reflect the presence of dew in the wheat canopy and at the soil surface during the morning, where acetic acid would be efficiently trapped. It may also be due to an afternoon source of acetic acid from the plant metabolism or soil that would partially compensate its deposition. Acid emission was indeed shown to increase with temperature (Filella et al., 2007) and transpiration (Kesselmeier et al., 1998). The diurnal mechanism of turgor regulation could also influence the net-acid exchanges by plants. During the day the pH level in the apoplasm are reduced, which increases the gaseous-to-aqueous ratio of acids and hence their volatilization (Gabriel et al., 1999).

Methylfuran ($m/z$ 83.049, $C_5H_6O$) was the sixth most deposited compound. It also had one of the highest deposition velocities in our study. Little can be found in the literature

**Table 2.** Fluxes and mixing ratios of the six most deposited VOCs[1] found in this study, compared to measurements reported in the literature over any type of vegetation.

| $m/z$ | Tentative raw formula | Tentative identification | Mixing ratio ppt | Flux nmol m$^{-2}$ s$^{-1}$ | $V_{dep}$ mm s$^{-1}$ | Reference |
|---|---|---|---|---|---|---|
| 75.044 | $C_3H_6O_2$ | Hydroxyacetone (HAC) | 510 [2–1200] [50–1300] | −0.05 [−0.2–0.02] −0.2 [−0.6–0.05] | −0.1 [−0.6–0.3] 1.4 ± 0.5 [0–2] | This study N2015 |
| 43.018 | $C_2H_2O$ | Fragments*, ethenone | 510 [50–1240] | −0.05 [−0.21–0.06] | 0.1 [−0.1–0.5] | This study |
| 61.028 | $C_2H_4O_2$ | Acetic acid | 220 [0–550] | −0.03 [−0.13–0.03] | 0.2 [−0.2–0.7] | This study R2011 CE7 |
| 83.049 | $C_5H_6O$ | Methylfuran | 40 [9–90] | −0.02 [−0.10–0.01] | −0.03 [−0.7–0.5] | This study |
| 47.013 | $CH_2O_2$ | Formic acid | 270 [0.8–740] | −0.08 [−0.3–0.02] | 0.3 [−0.2–1.0] | This study |
| 101.059 | $C_5H_8O_2$ | 4-Oxopentanal pentenoic acid | 20 [3–50] | −0.01 [−0.06–0.01] | 0.3 [−0.6–1.7] | This study R2011 CE8 |

* Formaldehyde was the most deposited compound measured. However because of the uncertainties linked with its response to water vapour, fluxes are not reported here. See Yáñez-Serrano (2021). N2015: Nguyen et al. (2015). Ranges were estimated from Fig. S17. CE9 R2011: Ruuskanen et al. (2011). Values in brackets are min and max. K1998: Kesselmeier et al. (1998). Fluxes were measured over barley and expressed per leaf area index. They were multiplied by the maximum area index observed in this study (7 m$^2$ m$^{-2}$) to convert them to comparable units. Values in square brackets are standard deviations.

about fluxes of this compound, but it is a very well-known oxidation product of isoprene (T. Karl et al., 2009). Helmig et al. (1998) reported Methylfuran mixing ratios of 90 ppt near the ground and 80 ppt throughout the whole boundary layer over a tropical forest, suggesting a near-ground source in this environment. Methylfuran is known to be formed primarily during combustion processes and is commonly detected in biomass burning plumes (Koss et al., 2018; Hatch et al., 2017). In this study the deposition velocity showed a marked diurnal cycle with a maximum around 14h and a minimum at night (Fig. 8). From week 3, a sharp increase in $m/z$ 83.049 deposition velocity was observed coinciding with the onset of senescence, as seen by the decrease in the net photosynthesis (Fig. 4). Diurnal and seasonal patterns were similar between $m/z$ 83.049 and $m/z$ 47.013 (formic acid) after week 3, suggesting a similar deposition process for the two ions during senescence (Figs. 6 and 8). Although unlikely, a possible change in the fragmentation patterns may also have contributed to the increase in methylfuran deposition in week 3.

4-Oxopentanal (4-OPA, $C_5H_8O_2$, $m/z$ 101.059) and also possibly pentenoic acid were reported to be emitted by pig slurry (Ni et al., 2012; Feilberg et al., 2015) as well as from cow and sheep farms (Kammer et al., 2020; Hobbs et al., 2004), but no deposition has been reported previously to our knowledge. Pentenoic acid is also known to play a role in bacteria–fungi interactions in soil (Scholler et al., 2002; Effmert et al., 2012). Jud et al. (2016) observed 4-OPA formation from ozonolysis of plant essential oils or diterpenes present on plant surfaces. Similarly, ozonolysis of squalene, a triterpene present in all plants, was reported to produce 4-OPA (Xiong et al., 2019). In this study, $C_5H_8O_2$ was behaving very similarly to methylfuran ($C_5H_6O$). Indeed, their mixing ratios had a correlation coefficient of 0.95, and both

showed similar deposition velocity patterns with an increase from week 3 onwards, suggesting that the deposition of these compounds was promoted during senescence.

### 4.3 VOC mixing ratios influenced by the nearby farm

Since many volatile organic compounds were bidirectionally exchanged, the question arises whether the observed mixing ratios correspond to those of the suburban region or were largely influenced by the local environment, especially the nearby farm with livestock and the surrounding fields. By analysing the mixing ratio frequency as a function of the wind direction and wind speed, several compounds were clearly identified as coming from the nearby animal farm (Figs. 9, S12–S14 and Table S3). Some of these VOCs are directly emitted by the farm (Kammer et al., 2020), namely acetic acid ($m/z$ 61.028), monoterpenes ($m/z$ 137.129 and its fragment $m/z$ 81.070) as well as $m/z$ 136.000, identified as a benzothiazole ($C_7H_5N_S$). The pollution rose of these compounds showed three typical behaviours (Fig. 9).

The VOC $m/z$ 135.117 (identified as $C_{10}H_{14}$, para-cymene) showed large mixing ratios for a narrow wind sector and low wind speed similar to the methane pollution rose (Fig. S12), which is a tracer of livestock. The observed diel variation in mixing ratios is typical for a local source that is emitting constantly during the day and night. Indeed, turbulent mixing and hence dilution were higher during the day than at night (Fig. S6). Compounds that behaved similarly were monoterpenes ($m/z$ 137.129) and an oxygenated monoterpene ($m/z$ 153.127) (Fig. S13).

By contrast, the VOC $m/z$ 89.042 ($C_4H_8O_2$) showed high mixing ratios for a larger wind sector and for both low and high wind speeds. This is a less usual behaviour, which may

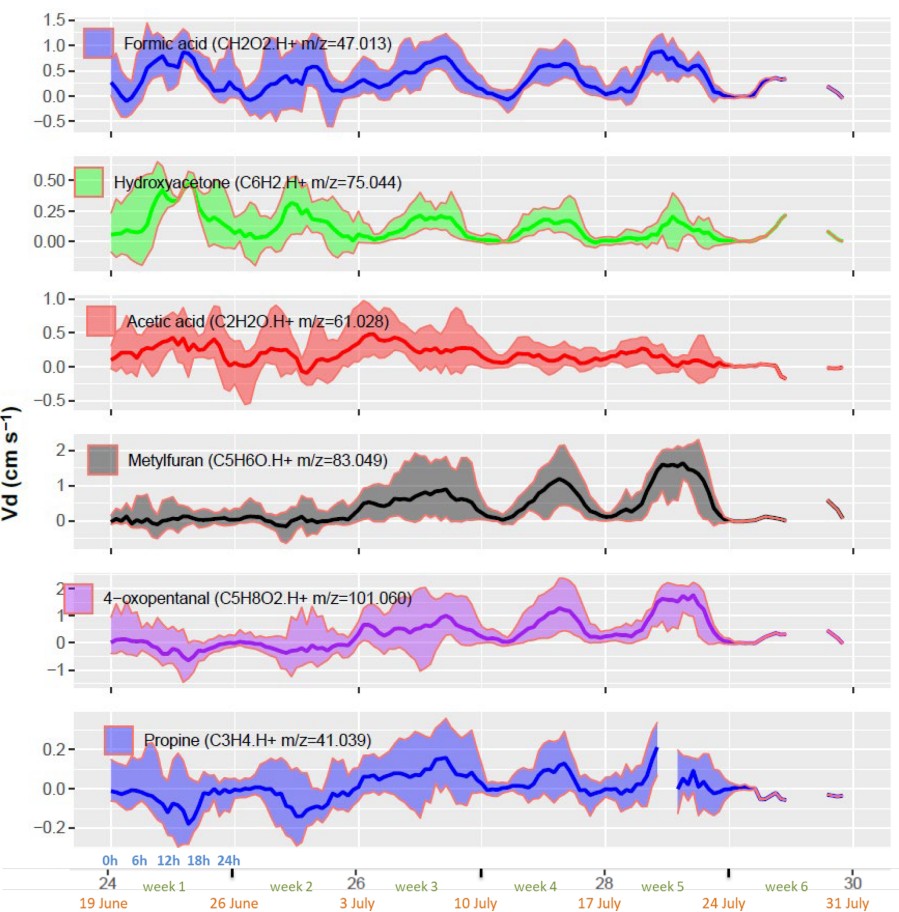

**Figure 8.** Diurnal cycle of deposition velocity of the six most deposited VOCs. Each week shows the diel cycle with its mean (line) and standard deviation (ribbons). The $x$ axis shows the week number in the year (black), the week number in the experiment (green), the starting date of the week (orange) and the hour of day (blue). Only 1 d of measurements was available during week 6, leading to no interquartile computation and hence no ribbons.

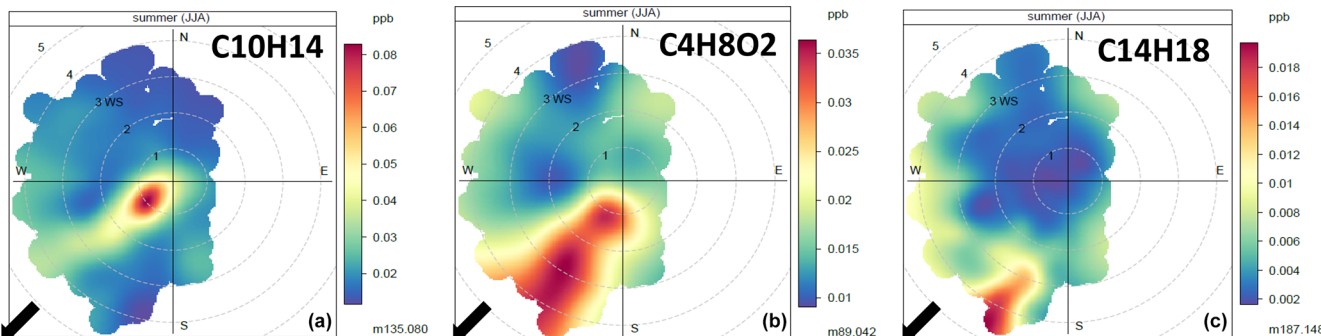

**Figure 9.** Three typical wind roses of compounds identified as coming from the farm. Colours show the mixing ratios, and plots are binned by averaged wind direction and wind speeds. **(a)** Constantly emitting source ($C_{10}H_{14}$). **(b)** Source emitting with a daily pattern ($C_4H_8O_2$). **(c)** Secondary photo-produced compound ($C_{14}H_{18}$). The arrows indicate the direction of the main farm buildings.

reflect the fact that emissions concomitantly increase with wind speed, hence compensating the turbulent mixing effect. Since wind speed and temperature are correlated with each other throughout the day (Fig. S6), we hypothesize that emissions of this compound increased with the temperature of the source inside the farm, which is influenced by the temperature outside. Indeed the mixing ratios for this compound scaled positively with air temperature (Fig. S13). VOCs behaving similarly were acetic acid ($m/z$ 61.028), $m/z$ 103.075 ($C_5H_{10}O_2$) and $m/z$ 117.084 ($C_6H_{12}O_2$). Kammer et al. (2020) found that all these compounds originate from the stable and are associated with silage feeding. Silage is indeed a source of these compounds as well as ethanol, linked to microorganism's activity which increase with temperature.

Finally, VOCs $m/z$ 187.148 ($C_{14}H_{18}$ or $C_{11}H_{22}O$) showed large mixing ratios for a narrow wind sector and only for wind speeds larger than $\sim 3\,\mathrm{m\,s^{-1}}$. Since these wind speeds occurred only in the middle of the day at that site (Fig. S6), we hypothesize that during their travel from the farm these compounds were either rapidly broken down and removed from the air and/or were secondarily produced from other primary emitted VOCs by photochemical reactions. Using the $3\,\mathrm{m\,s^{-1}}$ threshold observed in Fig. 9, we infer a travel time of $\sim 2.5$ min. Alternatively, these VOCs were transported from an incinerator 2 km away, which was identified as an $SO_2$ source (Loubet et al., 2012). This incinerator, which can emit at both day and night is however a bit further west than the farm ($225° N$). Therefore, the source of the VOCs representing $m/z$ 187.148 may rather be the farm. Compounds that behaved similarly were heavy ions ($m/z$ 201.139 and $m/z$ 201.233) indicating oxidized VOCs.

## 5 Conclusions

We found that the PTR-Qi-TOF-MS was sensitive enough to measure the fluxes of 264 VOCs by eddy covariance (77 emitting and 187 deposited, fluxes 3 times the $LOD_f$) above a wheat canopy during the ripening and senescence periods. The fact that fewer compounds were net-emitted than net-deposited can be explained by the higher atmospheric mixing ratios of the deposited compounds, compared to the emitting ones. In particular, the nearby farm contributed to increasing the ambient concentration of some VOCs, most of which were quite heavy, including monoterpenes, but also more light VOCs like acetic acid.

A detailed analysis of the eddy covariance flux computation revealed that the Webb–Pearman–Leuning (WPL) correction was negligible (most of the time lower than 2 %). Similarly, normalizing by the $H_3O^+$ cps at high frequency was shown to produce mostly negligible biases. Based on the water vapour cluster, we found a small high-frequency loss in our setup. However, we found it difficult to analyse the high-frequency response of the noisy signals of most VOCs, including methanol. Further analysis of the high-frequency losses with a PTR-TOF-MS is still required to better characterize these potential errors.

Our measurements confirm previous findings showing that methanol is the most emitted compound from wheat fields, representing 52 % of the total VOC moles emitted by the crop. Acetone and acetaldehyde were also found to contribute significantly to the summed VOC emissions. However, we detected several other compounds not previously reported as emitted by wheat, in particular, furan ($m/z$ 93.033), which may be released by decomposing organic matter. Moreover, substantial DMS emissions were measured, which confirms earlier findings on the magnitude of the terrestrial DMS source and suggests that soils may be a significant source of DMS together with plants and litter. Furthermore, we found large deposition rates of oxygenated VOCs, in particular formic and acetic acids together with hydroxyacetone and 4-oxopentanal. The deposition of many VOCs was enhanced during senescence, suggesting either an uptake process or a gas phase interaction with compounds emitted by the plant at that stage. These putative processes would need further investigations.

It is the first time 123 VOCs fluxes (move than 3 times $LOD_f$) measured by eddy covariance are reported over a wheat field. Overall, the summed VOCs emissions amounted to $41\,\mathrm{g\,ha^{-1}\,d^{-1}}$ and summed deposition amounted to -$26\,\mathrm{g\,ha^{-1}\,d^{-1}}$, leading to a net flux of $\sim 15\,\mathrm{g\,ha^{-1}\,d^{-1}}$. When roughly extrapolated to the whole year, the summed VOC flux amounted around $3\,\mathrm{kg\,C\,ha^{-1}\,yr^{-1}}$, an amount negligible compared to the $CO_2$ flux ($17\,\mathrm{Mg\,C\,ha^{-1}\,yr^{-1}}$). Our study confirms that the most emitted compound from wheat has a low reactivity (methanol). However, it highlights that wheat is an active sink for a number of oxygenated compounds.

Using a PTR-Qi-TOF-MS to measure VOC fluxes over an ecosystem for a long period by eddy covariance allows estimating the fluxes of a large number of compounds and establishing a net flux. The main difficulty continues to be the clear identification of the compounds corresponding to each ion as well as the calibration of a large number of gases.

**Data availability.** The complete hourly dataset is available as an Rdata file (COV3ER_2016_dataset.Rdata) containing the whole dataset, together with a data description file (COV3ER_2016_dataset-description.xlsx) and a script for reading and plotting the data (COV3ER_2016_dataset_make_graphs.R). The data and code for visualization are available at https://doi.org/10.15454/IRZ9XX (Loubet, 2021).

**Supplement.** The Supplement includes a PDF document with supplementary figures and tables and an Excel file containing the averaged fluxes, mixing ratios and $LOD_f$ (Table S3). Also available are three CSV files containing correlation tables between VOCs (Ta-

bles S4a, S4b, S4c). The supplement related to this article is available online at: https://doi.org/10.5194/acp-22-1-2022-supplement.

**Author contributions.** This study was conceptualized by BL, who also developed the software for data acquisition and analysis with OZ. The experiment was supervised by all authors except LA, JK, SB and MS. The data were curated by FL, RC, PB, LGG, BL, FT, DB and VG. The formal analysis was performed by BL, LGG, LA, FL, PB and RC. All authors except LA, JK, SB and MS provided resources. BL wrote the draft of the manuscript, and all authors contributed to the writing and reviewing. BL and VG contributed to finding the funding resources.

**Competing interests.** The contact author has declared that neither they nor their co-authors have any competing interests.

**Disclaimer.** Publisher's note: Copernicus Publications remains neutral with regard to jurisdictional claims in published maps and institutional affiliations.

**Acknowledgements.** We acknowledge funding by ADEME (COV3ER, no. 1562C0032) and ANAEE-FR services (ANR project no. 11-INBS-0001). The data analysis was supported by the regional funding R2DS and the ADEME projects DI-COV (grant no. 1662c0020) and AGRIMULTUPOL (grant number 1703C0012). Sandy Bsaibes acknowledges the European Union's Horizon 2020 research and innovation programme under the Marie Sklodowska Curie grant agreement no. 674911-IMPACT. The measurements were performed on the ICOS FR-GRI site. We acknowledge Dominique Tristan field site. The field site FR-GRI is part of the ICOS Europe and ICOS France infrastructures CE10.

**Financial support.** This research has been supported by the Agence de l'Environnement et de la Maîtrise de l'Energie (grant no. COV3ER, 1562C0032) and the Agence Nationale de la Recherche (grant no. ANAEE-FR, 11-INBS-0001).

**Review statement.** This paper was edited by Kyung-Eun Min and reviewed by two anonymous referees.

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

## Remarks from the language copy-editor

CE1    Please give an explanation of why this needs to be changed. We have to ask the handling editor for approval. Thanks.

CE2    Please give an explanation of why this needs to be changed. We have to ask the handling editor for approval. Thanks.

CE3    Please give an explanation of why this needs to be changed. We have to ask the handling editor for approval. The abbreviation is still used in the table. Thanks.

CE4    Please give an explanation of the requested additions and deletions here, as we have to ask the handling editor for approval. Thanks.

CE5    You have requested an addition here, but have also requested that the entire passage be deleted. Please clarify exactly how the notes under the table should read in the explanation for the editor. Thanks.

CE6    No problem. Thank you for double-checking.

CE7    Please give an explanation of why this needs to be changed. We have to ask the handling editor for approval. Thanks.

CE8    Please give an explanation of why this needs to be changed. We have to ask the handling editor for approval. Thanks.

CE9    Please give an explanation of the deletions here. We have to ask the handling editor for approval. Thanks.

CE10    Done.

## Remarks from the typesetter

TS1    Thank you for providing the link. A reference entry consists of: creator(s), title, link, last access and year. Please provide a reference entry.

TS2    The change differs from the other change. In this case. We would have to ask the editor for approval if you want to adjust it.

TS3    We need an explanation for the editor here as well.

TS4    Thank you for the clarification. Please note that a cdot is already used throughout so I do not see what else you need us to change. Sorry for the confusion in this matter.

TS5    Please confirm that the included reference is correct and add [code] or [data set] after the repository. The citation is used in the data section.