# Peer review of "Volatile organic compound fluxes over a winter wheat field by PTR-Qi-TOF-MS and eddy covariance"

_Atmospheric Chemistry and Physics, 2020_

## Author Comment (AC1)

**Answer to reviewer 1 comments**

**Reviewer 1 comments (https://doi.org/10.5194/acp-2020-1328-RC1)**

This manuscript presents eddy covariance flux measurements of VOCs over a wheat field in Europe with a PTR-Qi-TOF-MS. It identified several most emitted and deposited VOCs for a crop field, and presented their fluxes. The paper is a bit hard to follow. But studies for reactive species over croplands are rare. I still consider the paper provides valuable data to the community. The error characterization of eddy-covariance flux calculation is potentially helpful though they are mostly presented in the supplement.

We thank the reviewer for this comment. We have tried to improve the readability of the manuscript by reorganising the sections (as also suggested by the reviewer 2). In particular, we have reorganised the section on calibration to include a species-specific calibration. We have also shorten the results section on mixing ratios and the discussion session on fluxes. We have still left the eddy-covariance fluxes calculation details in the supplementary, as the manuscript is already long and especially the material and method section.

I have the following concerns and suggestions:

**Main concerns**

- Abstract: consider shortening it by cutting those introductions (1/3 of the current abstract) and highlight results. Thanks for the suggestion. We have shortened a bit this part.

- Sections 2.2.4 and 2.2.5 for Kinetic concentration and calibration: If I understand correctly, this study calibrated 5 VOCs, and the rest VOCs were estimated using the kinetic approach (Eqn. 1). This is a critical component of the paper, but unfortunately, it is confusing and lacks key details and discussions on measurement uncertainty. The kinetic approach using the same rate constant for all VOCs is subject to large uncertainties (maybe as high as +/-200% or more). Uncertainty and how that affects conclusions of the work should be discussed. What is the uncertainty for calibrated VOCs? How frequent was the calibration (what is 'several time' as mentioned)? How do these calibrations compare to each other? How do their calibration factors compare to if using the kinetic approach? This should give some range of error estimates, at least for these selected VOCs.

Thanks for this sound remark. We indeed used the kinetic approach, and are fully aware of its limitations, although at the time of the experiment we did not have the capability to do a better calibration. Following the reviewer suggestion, we have:

- (1) Modified the material and method section "2.2.5 Calibration procedures" which clarifies the calibration procedure. We have further added detailed part on the uncertainties computation for mixing ratios (with added material in the supplementary section). We have now included a calibration factor for each known compound based on a one-of calibration performed at the start of the experiment and calibration data from the work of Koss et al. (2018).
- (2) The discussion on the uncertainties in the calibration and how these may affect the conclusions of the work has been strengthen.

- Many figures in the texts are either not discussed or only very briefly discussed. Consider removing some of them as the manuscript is very long, and key messages would get lost with so much information blended in without actual contribution to the discussion.

Following the reviewer suggestion, we have moved Figures 4a, 4b that were only briefly used section 3.1 to supplementary material, while leaving Figure 5 that provides the most important parameters for fluxes.

- The figures and texts keep changing between using exact mass and species names. It needs to be consistent to improve the readability of the paper. I suggest using species names with exact masses in brackets if the mass is considered important.

We thank the reviewer for this remark and suggestion. We homogenised the notations in the text and figures according to the suggestion.

- Time series figures are interesting, but it took a while to figure out what they are and how to interpret them. I'd suggest either improving the presentation or captions to explain them.

Thanks for suggestion. We have tried to improve the readability by change the caption to "The x-axis shows the week number in the year (black), the week umber in the experiment (green), the starting date of the week (orange), and the hour of day (blue)."

- Section 3.2 VOC mixing ratio: it doesn't add much value, and the time series figure 6 is misleading to some extent (see above).

Considering the previous remark on the length and readability of the manuscript, we agree that these figures could be moved to the supplementary material, while keeping the text in the manuscript. This will help keeping the focus on the VOC fluxes and on the analysis of advected compounds from the nearby farm (in section 4.4).

- Figure 9: the deposition fluxes peaked around 5pm local time, which was explained as corresponding to afternoon rush hour. How is that possible since these are flux measurements which reflect a very small footprint given the height of the tower. Are there any traffics within the footprint? Further discussion on the footprint is needed. Later it presents the influence of the farm, and that should also be tied to the footprint discussion.

This is a sound remark.

- First of all, we should indeed stress that, yes the site is very close to high traffic roads especially close on the north-west (right at the edge of the field), but also on the east, and west and to a lesser extent on the south. This was discussed at length in Vuolo et al. (2017) about possible advection fluxes on NOx (section 3.5). The road on the north had a traffic load of 5000 to 15000 vehicles per day as a 2010 counts (Statistiques du département des Yvelines pour 2010), and the traffic has increased since then, especially on the east.
- Second, we should may be recall that deposition and emission are not related in the same way to the flux footprint. Indeed, assuming there is no bi-directionality in the flux for the sake of simplicity here, a deposition flux (F) can be expressed as the product of a deposition velocity (Vd) with a

concentration (C):  $F = V_d * C$ . Hence the peaks in deposition fluxes results from either both peaks in  $V_d$  or C. We can assume in a first order approach that  $V_d$  over a crop follows the same pattern as the friction velocity  $u_*$ , and will hence peak around noon in our site (Loubet et al., 2012; Vuolo et al., 2017). Hence, when we observe a peak in a deposition flux F around 5 pm it is most likely that this is due to variations C. To explain it in a different way, the deposition velocity is driven by the surface characteristics, and hence depends on the flux footprint, while the concentration depends on the concentration footprint, which is much larger than the flux footprint. Moreover, we should also bear in mind that concentration footprint concepts can be substituted by a concept of "air mass". At our site, we have clear evidence that the "air mass" changes when the wind is coming from the Parisian area (although Paris is at ~30 km away on the east). We usually observe an increase in NOx pollution when air is coming from the east (see Supp. Mat. Figures S2, in Vuolo et al. 2017). This is also, what we observed in this study for NO showing a clear increase from the east (Figure S12 reproduced below). In conclusion, it is very likely that some traffic emitted VOC concentrations follow a traffic-like pattern, which, if the compounds are deposited will drive the deposition flux pattern.

For the compounds advected from the farm, the same reasoning holds.

Figure S12. Pollution roses for CH4, NO and N2O.

- Table S2: I cannot find this table in the supplement. It states, 'See file Loubet-COV3ER-wheat-2016-EC-Supp.Mat-VF.', but this table of VOC tentative identification is not available. Is the 'Loubet-COV3ER-wheat-2016-EC-synthesis-fluxes-VF' what you referred to?

We are terribly sorry to have forgotten to include this file in the supplement. It is now included in the revised supplement.

- Section 4.2: Methanol. It is interesting that the study found that its emission increases towards senescence. Several studies have reported that methanol emission peaks when leaves are young during the early growing season, i.e., in the US (Karl et al., 2003; Hu et al., 2011), northern mid-latitude ecosystems (Wells et al., 2012), in MEGAN emission model (Guenther et al., 2012). Does this study suggest different mechanisms for crops/wheat to emit methanol compared to other ecosystems?

We were also surprised to find this behaviour for methanol fluxes. However, Gonzaga et al. (2021) also found this (Atm. Env. Accepted with minor revisions) in 2017 at the same site but with a different technique:

a chamber method, and by Bachy et al. (2020) with a similar technique. This suggest indeed that demethylation of pectin is still efficient during aging and senescence of plant tissues as shown by Michele et al. (1995), but also that another mechanism increases methanol fluxes. This may be due to a pronounced degradation of cellular constituents at the end of chlorosis that may promote methanol production and its release through higher leaf porosity chlorosis (Keskitalo et al., 2005; Woo et al., 2019). Increased methanol emissions have been reported above cut grasslands (Seco et al., 2007), which may suggest a similar process as during wheat senescence to occur in grasslands.

- Sections 4.2 and 4.3. Most of this section read like a literature review, and there is not much new knowledge added from what we already know about biogenic VOC emissions. I'd consider shortening them or likely merging with the previous sections when discussing their mixing ratio/flux results (Sections 3.3, 3.4, and 3.5.)

We thank the reviewer for the suggestion of shortening this section. We however disagree with the comment that there is not much benefit in this section. Indeed, we consider that we bring new results and new ideas in this section by comparing our results to the literature. As a first example, the previous comment shows that our results suggests a new methanol production process during plat senescence. Then, m/z 93.037 is for the first time reported to be emitted at that magnitude by an ecosystem. For acetaldehyde, our results suggests a new acetaldehyde emission route from crops through oxidation of organic matter by ozone that would be especially important during senescence. Regarding DMS, although our results confirm previous emission magnitudes, the discussion points towards a possible source from moist soil that we fill important to mention here. For monoterpenes, the discussion on the ion fragment suggested a dominating MT I our study (3-carene). Similarly, to m/z 93.037, phenols are first reported as being emitted by plants or soil.

Regarding VOC deposition, there is little references reporting deposition over crops and the discussion allowed positioning this study with respect to the existing literature.

To conclude, we have hence shorten this section as suggested by the reviewer, but kept it as a separate discussion where new results have been highlighted.

- Species 'tentatively attributed to': This term is overused in the paper. Sure, some compound identifications are tentative, but many others are certainly based on literature and the correlation analysis. A thorough and careful assessment of species identification is needed to address what is tentatively attributed.

We have simplified the text to get rid of this term in most sections except in when strictly necessary (m/z 93.037). A thorough and careful assessment of species identification was indeed performed based on the literature and the lab experience.

- Section 4.4: influence by the farm. Would the farm affect the flux measurements? Does the flux footprint cover the farm? It should be easy to quantify that influence since footprint analysis is already done.

Farm is not in the flux footprint but in the concentration footprint for sure (see answer to the previous comment on traffic). As already explained, since the  $F = V_d * C$ , if the concentration C is influenced by the farm then the flux F is also influenced by the farm. In addition, indeed, our data shown clearly that the farm influences the concentration of methane (Figure S12 reproduced above), and some VOCs (Figure 10 reproduced below). In these figures, it is clear that the methane and  $C_{10}H_{14}$  show enhanced concentrations

when the wind is blowing from the farm. Since the farm is the largest methane source in the surrounding, we can conclude that  $C_{10}H_{14}$  is also coming from the farm. We have clarified this in the manuscript.

Figure 10. Three typical wind rose of compounds identified as coming from the farm. Colours show the mixing ratios, and plots are binned by averaged wind direction and wind speeds. Left: constantly emitting source (example  $C_{10}H_{14}$ ). Middle: source emitting with a daily pattern (example of  $C_4H_8O_2$ ). Right: secondary photo-produced compound (example m/z 187.148, possibly  $C_{11}H_{22}O_2$ )

- Correlation analysis is helpful to identify possible fragmentation. The paper provides the results for two electric fields E/N 130 and 150. Do those excluded species show similar patterns under different E/N fields? Or do they fragmentation patterns change, at least for some masses?

Thanks for this sound question. Correlation results shown in the supplement data table S4b and Table S4c were used to find species that fragmented in both E/N fields. Overall, as expected, the fragmentation was much larger with E/N = 150 than in E/N = 130. E/N=150 also tend to give more C fragments, while E/N=130 lead to more H2O clusters. Three examples are given in the table below that shows that the pattern are quite similar between the two E/N. We see that isoprene clearly fragmented to m/z 57 and m/z 41 in similar amounts for the two E/N. Similarly, monoterpenes fragmented to m/z 81 in similar amounts for the two E/N but fragments to m/z 125 only with E/N = 150. C8H12 fragmented to a lot of m/z with E/N 150 but to only one significantly with E/N 130.

| m / z   | formula | m/z
missing
fragment | fragment
missing
formula | peak ratio @
E/N=150 | Corr. coeff.
@ E/N=150 | peak ratio
@ E/N=130 | Corr. coeff.
@ E/N=130 | Ratio of
peaks
150/130 |
|---------|---------|----------------------------|--------------------------------|-------------------------|---------------------------|-------------------------|---------------------------|------------------------------|
| 137.132 | C10H16  |                            |                                |                         |                           |                         |                           |                              |
| 125.132 | C9H16   | 12.00                      | С                              | 2.09                    | 0.95                      | NA                      | NA                        | NA                           |
| 81.070  | C6H8    | 56.06                      | C4H8                           | 2.71                    | 0.95                      | 2.47                    | 0.98                      | 1.10                         |
|         |         |                            |                                |                         |                           |                         |                           |                              |
| 69.070  | C5H8    |                            |                                |                         |                           |                         |                           |                              |
| 57.070  | C4H8    | 12.00                      | С                              | 1.63                    | 0.98                      | 1.37                    | 0.93                      | 1.19                         |
| 41.039  | C3H4    | 28.03                      | C2H4                           | 3.42                    | 0.97                      | 3.14                    | 0.81                      | 1.09                         |
|         |         |                            |                                |                         |                           |                         |                           |                              |
| 109.101 | C8H12   |                            |                                |                         |                           |                         |                           |                              |
| 95.088  | C7H10   |                            | CH2                            | 2.94                    | 0.95                      | 3.21                    | 0.95                      | 0.92                         |
| 83.085  | C6H10   | 26.02                      | C2H2                           | 1.85                    | 0.95                      | 1.78                    | 0.81                      | 1.04                         |
| 71.090  | C5H10   | 38.01                      | C3H2                           | 0.81                    | 0.95                      | 1.90                    | 0.80                      | 0.43                         |
| 69.070  | C5H8    | 40.03                      | C3H4                           | 2.38                    | 0.97                      | 4.83                    | 0.89                      | 0.49                         |
| 57.070  | C4H8    | 52.03                      | C4H4                           | 3.85                    | 0.97                      | 6.21                    | 0.88                      | 0.62                         |
| 41.039  | C3H4    | 68.06                      | C5H8                           | 8.33                    | 0.96                      | NA                      | NA                        | NA                           |

- There are many typos, minor grammatical errors, citation errors. Thorough proofreading could help.

Thanks for this comment. We have asked an English speaker to proofread the manuscript.

**Specific comments**

- Line 125: What is the purpose of the 16-way sulfinert coated valve?

The Sulfinert valve was used for switching to calibrations as well as vertical profiles and chamber measurements that are not reported in this manuscript. We however think it is better to give this detail here for a complete understanding of the setup.

- Line 144: What is teflonised pump? Explain it?

A teflonised pump is a pump which internal chamber is made of Teflon to avoid any interference. The reference has been added in the text for clarity sake.

- Section 2.2.1: what is the mass resolving power of this instrument? It'd be important for species identification. This is a sound remark. The mass resolving power corresponded to a resolution (ratio of ion peak width at mid-height to peak value) of around 4500 during the experiment. This means that the instrument had a mass resolving power of ~0.007 m/z at m/z = 30 and ~0.03 m/z at m/z = 150. The last two sentences have been added in the manuscript.

- Line 158: Lower the electric field to diminish cluster formation and fragmentation. Even 129 Td, there'd be lots of fragmentation. I'd reword it.

Thanks for the suggestion. We do agree. We changed the sentence to "The E/N ratio at the start of the experiment was rather high and was hence lowered down to 129 Td in the second half of the experiment to minimise cluster formation and fragmentation, although the latter cannot be avoid".

- Line 206: Why 'a single calibration factor for all VOC using toluene'? I thought you performed calibrations for 5 VOCs, no?

This has been changed completely, thanks to the useful reviewer comments. See answers to previous comments.

- Line 354 and Figure 5: the flux footprint is presented as unitless. It needs to define what the footprint is and how to interpret it.

The following explanation has been added in the manuscript: "The flux footprint  $\varphi(x, y)$  of the measurement mast is the probability density that the measured flux F originates from the field point of coordinates (x, y). The measured flux is then:  $F = \int \varphi(x, y) f_0(x, y) dx dy$ , where  $f_0(x, y)$  is the surface flux at coordinates (x, y). The flux footprint should be distinguished from concentration footprint h(x, y) that relies the concentration measured at the mast C minus the background concentration  $C_{bgd}$  to the surface flux:  $C - C_{bgd} = \int h(x, y) f_0(x, y) dx dy$ . Assuming  $f_0(x, y)$  is constant over the studied field, which is a reasonable assumption for a crop; the previous equations can be integrated to provide integrated footprints:

$$F = f_0 \times \int_{field} \varphi(x, y) dx dy = f_0 \times \emptyset_{field}$$
(7)

 $C - C_{bgd} = f_0 \times \int_{field} h(x, y) dx dy = f_0 \times H_{field}$

Here  $\phi_{field}$  has no units while  $H_{field}$  has units of a transfer resistance (s m-1)."

- Line 798: The sentence does not seem to be complete after 'New developments in this field would be helpful'?

Yes indeed. We replaced it by the following: "Further analysis of the high frequency losses using a PTR-TOF-MS is still required to better characterise these potential errors."

**References cited by the reviewers**

- Karl, T., Guenther, A., Spirig, C., Hansel, A., and Fall, R. (2003), Seasonal variation of biogenic VOC emissions above a mixed hardwood forest in northern Michigan, Geophys. Res. Lett., 30, 2186, doi:10.1029/2003GL018432, 23.
- Hu, L., Millet, D. B., Mohr, M. J., Wells, K. C., Griffis, T. J., and Helmig, D.: Sources and seasonality of atmospheric methanol based on tall tower measurements in the US Upper Midwest, Atmos. Chem. Phys., 11, 11145–11156, https://doi.org/10.5194/acp-11-11145-2011, 2011.
- Wells, K. C., Millet, D. B., Hu, L., Cady-Pereira, K. E., Xiao, Y., Shephard, M. W., Clerbaux, C. L., Clarisse, L., Coheur, P.-F., Apel, E. C., de Gouw, J., Warneke, C., Singh, H. B., Goldstein, A. H., and Sive, B. C.: Tropospheric methanol observations from space: retrieval evaluation and constraints on the seasonality of biogenic emissions, Atmos. Chem. Phys., 12, 5897–5912, https://doi.org/10.5194/acp-12-5897-2012, 2012.
- Guenther, A. B., Jiang, X., Heald, C. L., Sakulyanontvittaya, T., Duhl, T., Emmons, L. K., and Wang, X.: The Model of Emissions of Gases and Aerosols from Nature version 2.1 (MEGAN2.1): an extended and updated framework for modeling biogenic emissions, Geosci. Model Dev., 5, 1471–1492, https://doi.org/10.5194/gmd-5-1471-2012, 2012.

**References cited in the answers**

- Keskitalo, J., Bergquist, G., Gardestrom, P. and Jansson, S., 2005. A cellular timetable of autumn senescence. Plant Physiology, 139(4): 1635-1648.
- Koss, A.R. et al., 2018. Non-methane organic gas emissions from biomass burning: identification, quantification, and emission factors from PTR-ToF during the FIREX 2016 laboratory experiment. Atmospheric Chemistry and Physics, 18(5): 3299-3319.
- Loubet, B. et al., 2012. Investigating the stomatal, cuticular and soil ammonia fluxes over a growing tritical crop under high acidic loads. Biogeosciences, 9(4): 1537-1552.
- Michele, N.-M., MacDonald, R.C., Jennifer, J.F., Cheryl, L.W. and Fall, R., 1995. Methanol Emission from Leaves: Enzymatic Detection of Gas-Phase Methanol and Relation of Methanol Fluxes to Stomatal Conductance and Leaf Development. Plant Physiology, 108(4): 1359-1368.
- Seco, R., Penuelas, J. and Filella, I., 2007. Short-chain oxygenated VOCs: Emission and uptake by plants and atmospheric sources, sinks, and concentrations. Atmos. Environ., 41(12): 2477-2499.
- Vuolo, R.M. et al., 2017. Nitrogen oxides and ozone fluxes from an oilseed-rape management cycle: the influence of cattle slurry application. BIOGEOSCIENCES, 14(8): 2225-2244.
- Woo, H.R., Kim, H.J., Lim, P.O. and Nam, H.G., 2019. Leaf Senescence: Systems and Dynamics Aspects. In: S.S. Merchant (Editor), Annual Review of Plant Biology, Vol 70. Annual Review of Plant Biology, pp. 347-376.

---

## Author Comment (AC2)

**Answers to reviewer 2 comments**

**Reviewer 2 comments (https://doi.org/10.5194/acp-2020-1328-RC2)**

The authors present multi-week measurements of hundreds of VOC fluxes over a wheat field. They discuss the technical aspects of their measurement well, making this work quite useful for further investigations. They highlight the highest positive and negative flux compounds, and do a good job expounding upon their significance. Overall, I consider this paper valuable to the community. I have some concerns, listed below. Most importantly, please ensure that the compound assignments are correct and match the very impressive SI. Secondly, while the authors go into significant technical detail, there are still areas to improve clarity, and ensure future studies can benefit from the techniques used in this work.

**We thank the reviewer for his comments. We agree that the manuscript sill needed clarification in the compound assignment. This has been ow performed making better use of the fragment analysis, and also of the literature on PTR-TOF-MS calibrations as explained in our answers to reviewer 1.**

**Abstract:**

please state the start date and duration of the study in the abstract.

**Thanks for the sound suggestion. We also added the location and plant stages with the following sentence: "The study took place near Paris over a 5 weeks period starting the 3$^{rd}$ June 2016 spanning crop maturity and senescence."**

Mass 93.037 or 93.033? In the abstract and at line 802 this mass is referred to as mass 93.037, not 93.033. This is quite important: if the measured mass was 93.037, it may be more appropriately identified as $C_3H_9OS+$, or 2-Methylmercaptoethanol as identified in the GLOVOCS database. Additionally, this would put the assigned formula of $C_6H_5O+$ some 38 ppm from the measured mass, an error much larger than one expects from the authors' instrument.

**This is a very sound comment, which needs clarification. We have reprocessed a spectral analysis of this peak over some key hours for this compound. We also used the knowledge of the fragment and isotopes to get a more robust estimate and we are now convinced that this is actually an oxygenated compound ($C_6H_4O$, possibly furan), with a very small 0.5 ppm mass error. Other compounds have been reanalysed in a similar manner and Table S2, and figures have all been corrected accordingly.**

**Methods:**

Please explicitly state the number of days the experiment lasted.

**This has been included: 46 days**

Also, for the sake of our backwards American counterparts, please consider using an unambiguous date format in the text, such as June 3rd 2016 rather than 03/06/2016 (although the latter is perfectly fine in figures and tables).

**Thanks for the suggestion. We changed the date format in the text.**

Would it be possible to add the events discussed in lines 101-111 to Figure 3?

**Thanks for the suggestion. We have changed Figure 3 to include these. It shows like this now:**

[Figure]

**Figure 1. Top: evolution of the above ground biomass of different plant compartments. Bottom: crop height, crop developmental stages and farmer activity. The experimental period is highlighted in grey shading.**

Section 2.2.1: Can you provide the Reynolds number for the sampling line? Can you comment on the height of the tower, as it seems short relative to the height of the wheat. Does this make calculating a footprint difficult? Also please report the size of the footprint in this section.

**The Reynolds number for the sampling line was set to 6120, so well above the critical value of 2000-3000, to ensure a turbulent regime to avoid high frequency losses. The Reynolds number value was added in the text. The crop heights $h$ was below 1 m and the mast height was at 2.7 m, which gives a mast height above the displacement height d (~ 0.7h) larger than 2 m. This is within the standard eddy-covariance measurement heights for low ecosystem heights such as crops. Rebmann et al. (2018) recommended a measurement height between 1.6 and 6 times the canopy height, which corresponds to our setup. 3 m might have been better of course, but we also wanted to minimise the risk that the footprint goes out of the field, especially for south and east wind directions, given the specific setup with a 30 m line as shown in Figure 2 and 3.**

**This 2.7 m height tower does not make the footprint calculation difficult as again this is a standard height for crops and the FIDES footprint model (Carozzi et al., 2013; Loubet et al., 2018) and others (Wilson et al., 2012) have been applied extensively for these heights?**

**We did not report the footprint in this introductive section but rather used a flux footprint model to compute the footprint and reported it as a result in section 3.1 (Figure4): "The flux footprint from the main field was mostly above 0.8 (median 0.86, interquartile 0.76 - 0.91) but showed some consistent periods with a lower footprint (down to 0.4) when the wind was blowing from the south. The periods with a footprint lower than 0.6 occupied 13% of the time."**

Section 2.2.4: This section feels a bit rushed, and as this is an AMT paper it would be appropriate to walk the reader through these steps. Additionally, along with section 2.2.5 this would be a good place to discuss uncertainty in both mixing ratio and flux measurements. While many have used the default reaction rate constant to calculate transmission based mixing ratios, the method does have a substantial error stemming from the variability in rate constants. Regarding section 2.2.5, it seems the authors take the mixing ratio calculated in equation 1 and then correct it with a calibration factor derived from the instrument response to toluene. What exactly does this calibration correct for, and why are the mixing ratios calculated using the default reaction rate constant of 2.5e-9 in need of correction, as they are already normalized to the primary ion signal, accounting for MCP and other changes?

**This is a very sound comment, which was very consistent to reviewer 1 major comment. We have modified our approach to account for variations in *k*. Please see reviewer 1 section detailed answers to this key point.**

In the methods section, please report the amount of time for which you were able to calculate fluxes. For times when you could not, did you do any gap-filling, and if so what was your method.

**We were able to calculate the flux for 85% of the time. We did not do any gap filling on the VOC fluxes, as this would require a well-validated VOC flux model. It was furthermore not the primary objective of this study to provide time integrated VOC fluxes, but rather to provide high quality VOC fluxes for modelling purpose. We have reported this in the method section.**

**Results:**

For figures 4, 5, 6, 7, 8, the standard deviation ribbons disappear in the final weeks of the plots. Why is this? Also, please add the assigned formula or compounds to the plots, not all of us know that m59.049 is acetone.

**The reason for the missing ribbons in the last week is that during this last week, there was only one day of measurement, hence leading to no estimate of the interquartile. We have mentioned this in the Figure legends when appropriate (Figure 6 and 7 of revised manuscript)**
**The second comment was also made by reviewer 1. All chemical species formula and names have been added to the figures when appropriate.**

Line 366: "The most concentrated VOC at the site were methanol, acetone, C6H4O, propanoic acid, ketene, propyne, acetaldehyde, formaldehyde, and hydrazine acetate (Table S2)" This does not align with the data in table S2. For instance, hydrazine acetate is not listed. Please correct.

**Thanks for this very sound comment indeed. The VOC names in the text have been thoroughly checked and aligned to Table S2 that has been also completely checked, to answer previous comments from both reviewers regarding VOC identification.**

In line 485, the authors mention butene (m/z 57.070), but it does not appear in table S2. Please ensure that Table S2 and the main text are in agreement on both names, formulas, and masses. If a compound is discussed in the

main text, I would like to see it included in S2. Additionally, have the authors considered that butanol will likely fragment onto the butene assigned ion?

**This comment feeds into the other comments on fragments and compound identification. We have reviewed our methodology and have accounted for these fragments now.**

The authors present a measurement of formaldehyde, which shows a high humidity dependance in PTR due it having a similar proton affinity as water. Can the authors show some figures in the SI that show their formaldehyde measurement is not too influenced by water vapor concentration? Otherwise I would not report the value, as it is not too discussed and there are many other interesting findings.

**Formaldehyde was indeed shown to depend much on humidity. We however do not have specific measurements with this instrument to allow evaluating and correcting for humidity effects on formaldehyde. When we plot the mixing ratio (MR, ppb) and the flux (nmol m-2 s-1) of formaldehyde against the vapour pressure pvap (kPa), we do not observe any clear trend (see figure below). However, this does not exclude that there might be an influence of water vapour on formaldehyde measurement. We have followed the reviewer suggestion not to report the formaldehyde fluxes and mixing ratios in Tables.**

**Section 3.5: Very interesting!**

**Discussion:**

Large emissions of MeOH have been seen from dairy operations. Could methanol be coming from runoff from the nearby animals?

**This is not plausible as the field is at 600 m from the farm (Figure 1). Moreover, the animals are mostly in buildings were the faeces are collected and stored for field applications. Cows go outside the farm but in an even further location on the west and south. Finally, the topography would not allow runoff to the field.**

Table 1: please explain the format of the flux better. I am a bit confused by the table note "Mean ± se [5 – 95 percentiles] and max – min" and how it relates to the fluxes. Also, when using the tilde, "~", you seem to omit the negative sign, which could lead to confusion. For instance, the first flux column for monoterpenes is negative, but the second reads positive.

**Thanks for the comment. Indeed, there should be a minus sign in the second column for monoterpenes but also for isoprene. To clarify we have deleted the tilde in the table and the percentiles, which were only given for Bachy et al. and not compared to this study. We only kept the mean ± standard error and the [min – max] ranges.**

Is it possible that the signal at 68.06 is from O2+ ionization of isoprene? If so were O2+ levels stable?

**Yes, this is the hypothesis we made here. Yes, the O2 levels were pretty stable at around 5%. See figure below that has been added in the SI, and shows the percentage of O2+ to H3O+ measured**

[Figure]

Lines 621-637: This is very good analysis.

**Thanks for this kind comment.**

4-OPA is missing from Figure 10, please include it. 4-OPA is a known ozonolysis product of squalene a component of human skin oil, and sometimes cited as a tracer for skin oil ozonolysis. It is very interesting to see it emitted/deposited in a wheat field. Is it known if wheat produces squalene as well? Did the authors see a pattern with 4-OPA and ozone?

**This is an interesting comment. Squalene is indeed a metabolite present in all plants playing a key role in the cell membrane properties (Lozano-Grande et al., 2018). It is also produced by wheat, and present in the grain, though I small quantities (~4 µg g⁻¹) (Konopka et al., 2017). It is mostly present in grains that contains lipids and amaranth is by far the plant containing the largest quantities (8% of its oil). In crops it is most present in soybean maize and sunflower oils (Lozano-Grande et al., 2018). The observed trend in 5-OPA mixing ratio would be compatible with the growth of maize in the surrounding sites (maize most intense growth occurs end of June-July).**

**It is striking here to see that 4-OPA emissions increased during senescence, which is also the grain-filling period where metabolites are transferred in the plant to the grain.**

**During weeks 26-28, 4-OPA mixing ratio (MR, ppb) was strongly correlated to the ozone concentration (O3, ppb), while the deposition velocity is primarily correlated to u* (Ustar, m s⁻¹), as shown in the figure below. The latter is expected from turbulent transfer theory. However, before senescence, the field did not show any peculiar deposition of 4-OPA. The link between 4mixing ratio and ozone concentration seem to suggest that 4-OPA at the field site resulted from an ozonation process. However, we measured a deposition and not an emission processes over wheat, indicating that the source elsewhere.**

[Figure]

**Figure. 4-OPA (m/z 101.060) mixing ratio (MR, ppb) as a function of ozone concentration (O3, ppb), and deposition velocity (Vexch) as a function of u* (Ustar, m s⁻¹). A boxplot shows the difference in deposition velocity before and after senescence.**

Figure 11: consider adding the direction of the farm to these plots.
**Thanks for this suggestion. We have done so.**

**Minor issues:**

Line 29: "outmost"?
**Changed to "high"**

Line 95: "the field is at around…" remove "at"
**Changed to "The field is around 450 m downwind from the farm buildings…"**

Line 93: "The site that is part of a dairy farm receives a lot of nitrogen as mineral or organic matter, which leads to large ammonia emissions" Are you referring to manure and runoff?
**No we are referring to volatilisation. We clarified to "The site that is part of a dairy farm receives a lot of nitrogen as mineral or organic matter, which leads to large ammonia volatilisation to the atmosphere…"**

Equation 1: Are there units for this constant?

This constant has indeed some units (L mbar$^2$ V$^{-1}$ s$^{-1}$ mol$^{-1}$ K$^{-2}$). When multiplied by $\frac{U_{drift}\,T_{drift}^2}{k\,p_{drift}^2}$ is has units of (L L$^{-1}$) which is equivalent to mol mol$^{-1}$.

Line 223: While I like the idea of a perfect gas constant, I believe you mean "ideal"

Yes of course! Thanks for spotting this mistranslation from the French name "loi des gaz parfaits". It has been corrected.

Line 288: "7 NL per min" I'm unfamiliar with "NL"

NL stands for Litre inn standard conditions. It has been explained in the manuscript.

Line 387: "Region" to "region"

Done

Line 467-68: please rephrase

We rephrased the last two sentences to "We showed that the bias is however small and negligible when integrated over time (Figure S2 and S3). Nevertheless, we recommend to calculate the covariances using raw cps and normalise them by the primary ion H$_3$O$^+$ afterwards, to avoid this minimal though proven bias. This is important especially in conditions with very strong fluxes since $\overline{w'\,cps'_{H_3O^+}}$ may increase under such conditions."

Line 564: extra "."

Indeed, there was a missing part in this sentence. It has been rephrased as ". Bachy et al. (2020) found acetone to behave very similarly to acetaldehyde…"

Line 673: "leaves" to "leaf"

Changed

Line 682: "under brackets" to "in brackets"

Changed

Line 792: "less" to "fewer"

Changed

**References cited in the answers**

Bachy, A. et al., 2020. Dynamics and mechanisms of volatile organic compound exchanges in a winter wheat field. Atmos. Environ., 221.

Carozzi, M., Loubet, B., Acutis, M., Rana, G. and Ferrara, R.M., 2013. Inverse dispersion modelling highlights the efficiency of slurry injection to reduce ammonia losses by agriculture in the Po Valley (Italy). Agric. For. Meteorol., 171: 306-318.

Konopka, I. et al., 2017. Variation of wheat grain lipid fraction and its antioxidative status under the impact of delayed sowing. Journal of Cereal Science, 76: 56-63.

Loubet, B. et al., 2018. Evaluation of a new inference method for estimating ammonia volatilisation from multiple agronomic plots. Biogeosciences, 15(11): 3439-3460.

Lozano-Grande, M.A., Gorinstein, S., Espitia-Rangel, E., Davila-Ortiz, G. and Martinez-Ayala, A.L., 2018. Plant Sources, Extraction Methods, and Uses of Squalene. International Journal of Agronomy, 2018.

Rebmann, C. et al., 2018. ICOS eddy covariance flux-station site setup: A review. International Agrophysics, 32(4): 471--494.

Wilson, J.D., Flesch, T.K. and Crenna, B.P., 2012. Estimating Surface-Air Gas Fluxes by Inverse Dispersion Using a Backward Lagrangian Stochastic Trajectory Model. In: J. Lin et al. (Editors), Lagrangian Modeling of the Atmosphere. Geophysical Monograph Series, pp. 149-161.